# Closed Knight's Tours on $(m, n, r)$-Ringboards

**Wasupol Srichote** [1], **Ratinan Boonklurb** [1,*] and **Sirirat Singhun** [2]

[1] Department of Mathematics and Computer Science, Faculty of Science, Chulalongkorn University, Bangkok 10330, Thailand; Wasupol.Sr@student.chula.ac.th
[2] Department of Mathematics, Ramkhamhaeng University, Bangkok 10240, Thailand; sin_sirirat@ru.ac.th
* Correspondence: ratinan.b@chula.ac.th

**Abstract:** A (legal) knight's move is the result of moving the knight two squares horizontally or vertically on the board and then turning and moving one square in the perpendicular direction. A closed knight's tour is a knight's move that visits every square on a given chessboard exactly once and returns to its start square. A closed knight's tour and its variations are studied widely over the rectangular chessboard or a three-dimensional rectangular box. For $m, n > 2r$, an $(m, n, r)$-ringboard or $(m, n, r)$-annulus-board is defined to be an $m \times n$ chessboard with the middle part missing and the rim contains $r$ rows and $r$ columns. In this paper, we obtain that a $(m, n, r)$-ringboard with $m, n \geq 3$ and $m, n > 2r$ has a closed knight's tour if and only if (a) $m = n = 3$ and $r = 1$ or (b) $m, n \geq 7$ and $r \geq 3$. If a closed knight's tour on an $(m, n, r)$-ringboard exists, then it has symmetries along two diagonals.

**Keywords:** legal knight's move; closed knight's tour; open knight's tour; Hamiltonian cycle; ringboard; annulus-board

## 1. Introduction

The $m \times n$ chessboard or $CB(m \times n)$ is the generalization of the regular $CB(8 \times 8)$. It consists of $m$ rows of $n$ arrays of squares. Suppose the squares of the $CB(m \times n)$ are labeled by $(i, j)$ in the matrix fashion. A legal knight's move is the result of a moving the knight two squares horizontally or vertically on the $CB(m \times n)$ and then turning and moving one square in the perpendicular direction. That is, if we start at $(i, j)$, then the knight can move to one of eight squares: $(i \pm 2, j \pm 1)$ or $(i \pm 1, j \pm 2)$ (if exists).

A closed knight's tour (CKT) is a legal knight's move that visits every square on a given chessboard exactly once and returns to its start square. While, an open knight's tour (OKT) is a legal knight's move that visits every square on a given chessboard exactly once and the starting and terminating squares are different. Both CKT and OKT problems on a two-dimensional or three-dimensional chessboard are one of the interesting mathematical problems as you can see some of them listed in [1–6]. Not only the legal knight's move, but some researchers also extended it to be an $(a, b)$-knight's move which is the result of a moving the knight $a$ squares horizontally or vertically on the $CB(m \times n)$ and then turning and moving $b$ squares in the perpendicular direction. Several mathematical problems along this direction were considered, see for examples [7–9] and references therein for details.

In 1991, Schwenk [10] obtained necessary and sufficient conditions for the existence of a CKT for the CB $(m \times n)$ as follows.

**Theorem 1.** ([10]) *A CB$(m \times n)$ with $m \leq n$ admits a CKT unless one or more of the following conditions holds: (i) mn is odd or (ii) $m \in \{1, 2, 4\}$ or (iii) $m = 3$ and $n \in \{4, 6, 8\}$. Furthermore, this CKT contains a knight's move from square $(1, n - 1)$ to square $(3, n)$ and square $(m, 2)$ to square $(m - 1, 4)$.*

For the CB($m \times n$) that contains no CKTs, DeMaio and Hippchen [11] and Bullington et al. [12] can provide the minimal number of squares to be removed or to be added in order for the obtained new board to have a CKT. In particular, for $m = 3$ or $m$ and $n$ are odd, Miller and Farnsworth [13] and Bi el al. [14] provided the exact position of a square to be removed from CB($m \times n$) so that the remaining board admits a CKT. However, for the case $m = 4$, the exact positions for two squares to be removed still open for researchers to explore.

In 2005, Chia and Ong [9] obtained necessary and sufficient conditions for the existence of an OKT for the CB($m \times n$) as follows.

**Theorem 2.** ([9]) *A CB($m \times n$) with $m \leq n$ admits an OKT unless one or more of the following conditions holds: (i) $m \in \{1,2\}$ or (ii) $m = 3$ and $n \in \{3,5,6\}$ or (iii) $m = 4$ and $n = 4$.*

In this article, we consider one of the variations of the CKT problem by considering the chessboard that the middle part is missing which is called $(m,n,r)$-ringboard or $(m,n,r)$-annulus board and we denote it by RB($m,n,r$).

**Definition 1.** *Let $m$, $n$ and $r$ be integers such that $m, n > 2r$. An RB($m,n,r$) is defined to be a CB($m \times n$) with the middle part missing and the rim containing exactly $r$ rows and $r$ columns.*

In 1996, Wiitala [15] showed that the RB($m,m,2$) contains no CKT. However, the characterization of the general RB($m,n,r$) has not been given. Thus, we try to establish the characterization like the one given by Schwenk [10]. Actually, the CKT problem on the RB($m,n,r$) can be converted to a certain graph problem. If we regard each square of the RB($m,n,r$) as a vertex, then a knight graph $G(m,n,r)$ represented all legal knight's moves on RB($m,n,r$) is a graph with $2r(m + n - 2r)$ vertices and two vertices $(a,b)$ and $(c,d)$ are joined by an edge whenever the knight can be moved from one square to another by a legal knight's move and this edge is denoted by $(a,b) - (c,d)$. Then, a CKT (respectively, OKT) on the RB($m,n,r$) is a Hamiltonian cycle (respectively, Hamiltonian path) in $G(m,n,r)$. The following theorem is a necessary condition for the existence of a Hamiltonian path in a graph that we often use in this article.

**Theorem 3.** ([9]) *Let $S$ be a proper subset of the vertex set of a graph $G$. If $G$ contains a Hamiltonian path, then $\omega(G - S) \leq |S| + 1$, where $\omega(G - S)$ is the number of components in $G - S$.*

The goal of this article is to prove that for $m, n \geq 3$ and $m, n > 2r$, the RB($m,n,r$) admits a closed knight's tour if and only if (a) $m = n = 3$ and $r = 1$ or (b) $m, n \geq 7$ and $r \geq 3$. In order to reach our goal, we need to divide our RB($m,n,r$) into small pieces depending on $r$. If $r \geq 5$ is even, then RB($m,n,r$) is divided into four smaller rectangular chessboard and we can use Theorem 1 to construct the CKT for RB($m,n,r$) which will be elaborated in Case 3.1 of Theorem 8 in Section 4. However, if $r \geq 5$ is odd and RB($m,n,r$) is divided into four smaller rectangular chessboards, then there is a case that Theorem 1 cannot be used (Case 3.2 of Theroem 8). Thus, we need to construct our own CKT base on the existence of an OKT on some rectangular chessboards which will be constructed in Theorem 6 in Section 3. For small $r$, namely $r \in \{3,4\}$, we need to divide RB($m,n,r$) into two parts, namely an L-board and a 7-board of widths 3 or 4 which we denote by LB($r,c,3$), LB($r,c,4$), 7B($r,c,3$) and 7B($r,c,4$) depending on the numbers of row $r$ and columns $c$ (See Cases 1 and 2 of Theorem 8). For example, Figure 1 illustrates that RB($10,11,3$) is divided into LB($10,8,3$) and 7B($10,8,3$) and RB($11,13,4$) is divided into LB($11,9,4$) and 7B($11,9,4$).

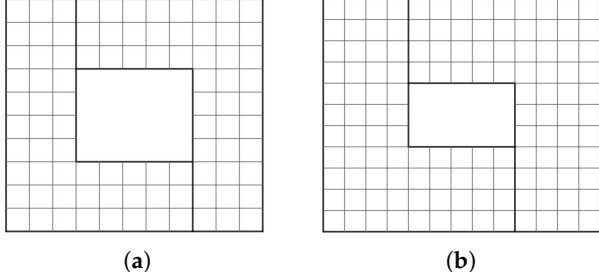

**Figure 1.** LB$(10, 8, 3)$, 7B$(10, 8, 3)$, LB$(11, 9, 4)$ and 7B$(11, 9, 4)$.

Therefore, to construct the CKT on the ringboard for this case, we prove the existence of a CKT on LB$(m, n, 4)$ and 7B$(m, n, 4)$ and the existence of some OKTs on LB$(m, n, 3)$ and 7B$(m, n, 3)$ are given in Theorems 4 and 5 in Section 2. For $r = 2$, we prove the extension of Wiitala's result in [15] which is the non-existence of the CKT on the RB$(m, n, 2)$ in Theorem 7 in Section 4. Finally, the conclusion and discussion about our future research are in Section 5.

## 2. CKTs and OKTs on Some LBs and 7Bs

First, let us construct the CKT on LB$(m, n, 4)$, where $m, n \geq 5$.

**Theorem 4.** *An LB$(m, n, 4)$ has a CKT containing an edge $(1, 4) - (3, 3)$ for all $m, n \geq 5$.*

**Proof.** First, let us construct CKTs on some small size LBs of the same and different parity of $m$ and $n$ as in Figures 2–4.

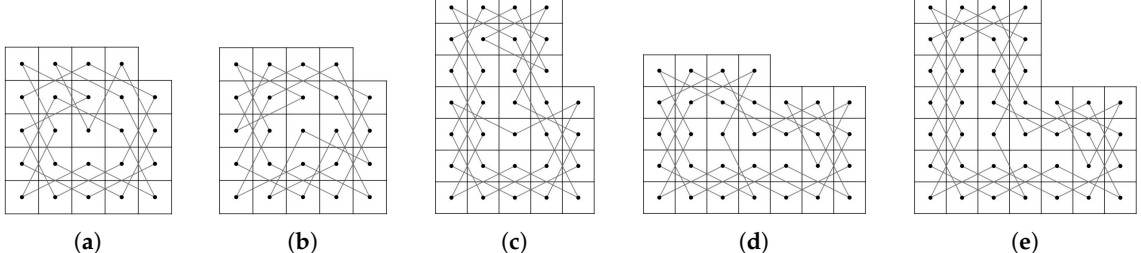

**Figure 2.** Closed knight's tours (CKTs) for LB$(5, 5, 4)$, LB$(7, 5, 4)$, LB$(5, 7, 4)$ and LB$(7, 7, 4)$.

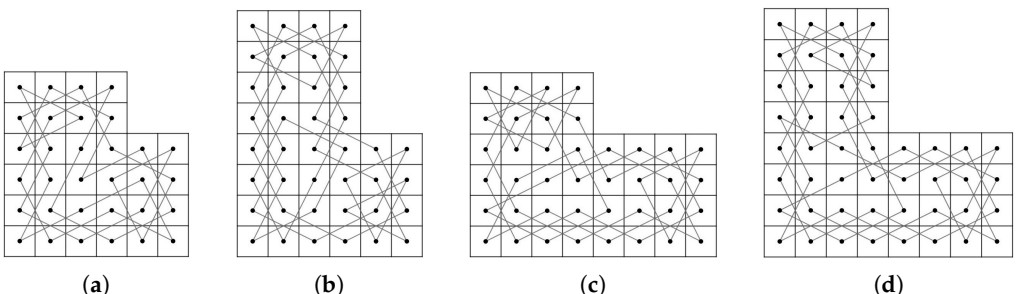

**Figure 3.** CKTs for LB$(6, 6, 4)$, LB$(8, 6, 4)$, LB$(6, 8, 4)$ and LB$(8, 8, 4)$.

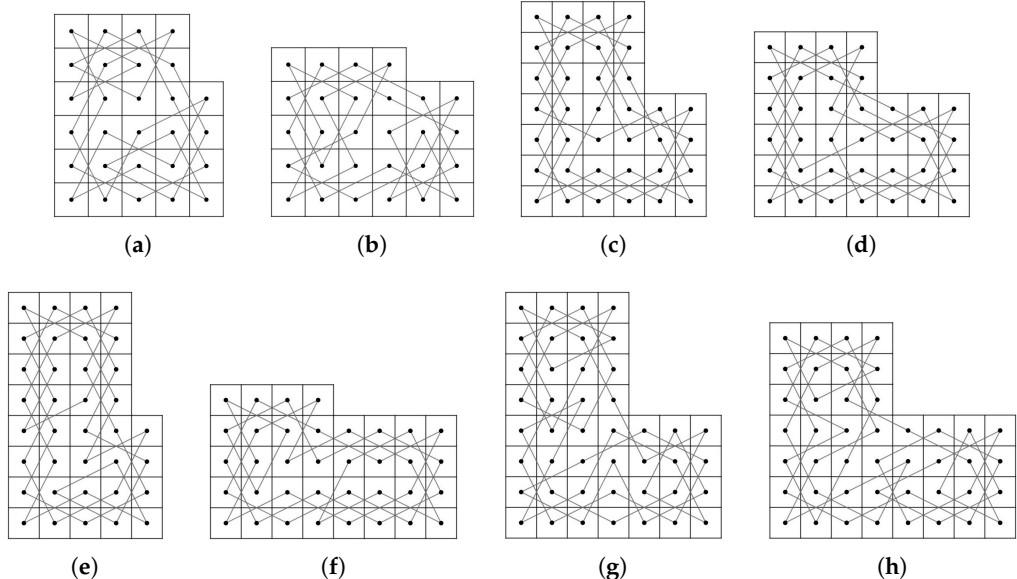

**Figure 4.** CKTs for LB$(6,5,4)$, LB$(5,6,4)$, LB$(7,6,4)$, LB$(6,7,4)$, LB$(8,5,4)$, LB$(5,8,4)$, LB$(8,7,4)$ and LB$(7,8,4)$.

Next, for the larger LBs, we start by constructing two paths $P_1$ (dash line) and $P_2$ (solid line) on the CB$(4 \times 4)$ as shown in Figures 5.

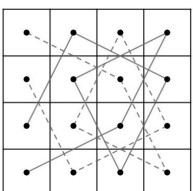

**Figure 5.** Two paths $P_1$ and $P_2$ on the CB$(4 \times 4)$.

Then, we construct two paths $P_1'$ and $P_2'$ on the CB$(4 \times 4t)$ where $t \geq 2$. Let us connect $t$ CB$(4 \times 4)$'s in Figure 5 to the right of each other and do the following.

(i)    For $1 \leq i \leq t - 1$, delete $(2,3) - (4,4)$ from $P_1$ and $(1,4) - (3,3)$ from $P_2$ of the $i$th CB$(4 \times 4)$.

(ii)   For $1 \leq i \leq t - 1$, join $(2,3)$ and $(4,4)$ of the $i$th CB$(4 \times 4)$ to $(1,1)$ and $(2,1)$ of the $(i+1)$th CB$(4 \times 4)$, respectively

(iii)  For $1 \leq i \leq t - 1$, join $(1,4)$ and $(3,3)$ of the $i$th CB$(4 \times 4)$ to $(3,1)$ and $(4,1)$ of the $(i+1)$th CB$(4 \times 4)$, respectively.

Notice that $(1,1)$, $(2,1)$, $(3,1)$ and $(4,1)$ are four end-points of two paths of the CB$(4 \times 4t)$ for $t \geq 1$. By rotating Figures 5 and 6 counter-clockwise by 90 degrees, we also obtain two paths $P_1''$ and $P_2''$ on the CB$(4s \times 4)$ where $s \geq 1$ as shown in Figure 7. Notice also that $(4s,1)$, $(4s,2)$, $(4s,3)$ and $(4s,4)$ are four end-points of two paths and the edge $(1,4) - (3,3)$ contained in one path of the CB$(4s \times 4)$ for $s \geq 1$.

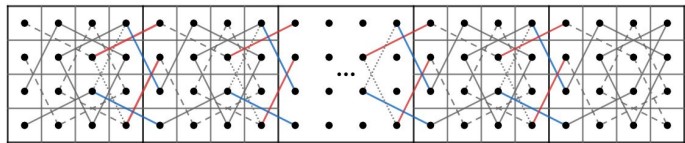

**Figure 6.** Two paths $P_1'$ and $P_2'$ on the CB$(4 \times 4t)$.

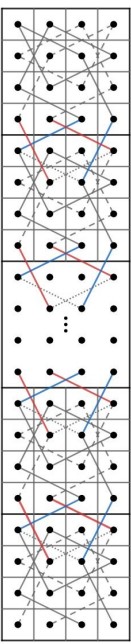

**Figure 7.** Two paths $P_1''$ and $P_2''$ on the CB($4s \times 4$).

Now, we are ready to construct a CKT on a larger LB by placing the CB($4 \times 4t$) to the right and the CB($4s \times 4$) above each smaller LB that we have considered before, respectively. WLOG, let $m \geq n \geq 5$.

- If $m$ and $n$ are odd integers, then $m \equiv 1$ or $3 \pmod{4}$ and $n \equiv 1$ or $3 \pmod{4}$.
- If $m$ and $n$ are even integers, then $m \equiv 0$ or $2 \pmod{4}$ and $n \equiv 0$ or $2 \pmod{4}$.
- If $m$ and $n$ are different parity, then $m \equiv 1$ or $3 \pmod{4}$ and $n \equiv 0$ or $2 \pmod{4}$; and $m \equiv 0$ or $2 \pmod{4}$ and $n \equiv 1$ or $3 \pmod{4}$.

Recall that the LB($a, b, 4$) has a CKT for all $a, b \in \{5, 6, 7, 8\}$. In addition, from Figures 2b–e, 3 and 4, each CKT of the LB($a, b, 4$) contains edges $(1, 1) - (2, 3)$, $(1, 4) - (2, 2)$, $(a - 3, b) - (a - 1, b - 1)$ and $(a - 2, b - 1) - (a, b)$. Furthermore, from Figures 2a,c–e, 3 and 4, each CKT of the LB($a, b, 4$) contains the edge $(1, 4) - (3, 3)$.

Thus, it is enough to show that the LB($a + 4s, b + 4t, 4$) has a CKT for any nonnegative $s, t$ and $a, b \in \{5, 6, 7, 8\}$ such that $s \geq t$ and $s \neq 0$. First, if $t = 0$, then let us divide the LB($a + 4s, b, 4$) into two subboards, CB($4s \times 4$) and LB($a, b, 4$). Otherwise, we divide into three subboards, CB($4s \times 4$), LB($a, b, 4$) and CB($4 \times 4t$). Then, we construct the required CKT by the followings.

(i) if $t = 0$, then delete $(1, 1) - (2, 3)$ and $(1, 4) - (2, 2)$ from the CKT of the LB($a, b, 4$). If $t > 0$, then further delete $(a - 3, b) - (a - 1, b - 1)$ and $(a - 2, b - 1) - (a, b)$ from the CKT of the LB($a, b, 4$).

(ii) If $t = 0$, then join $(4s, 1)$, $(4s, 2)$, $(4s, 3)$ and $(4s, 4)$ which are four end-points of two paths of the CB($4s \times 4$) to $(2, 2)$, $(1, 4)$, $(1, 1)$ and $(2, 3)$ of the LB($a, b, 4$), respectively. If $t > 0$, then further join $(1, 1)$, $(2, 1)$, $(3, 1)$ and $(4, 1)$ which are four end-points of two paths of the CB($4 \times 4t$) to $(a - 2, b - 1)$, $(a, b)$, $(a - 3, b)$ and $(a - 1, b - 1)$ of the LB($a, b, 4$), respectively .

Figure 8 illustrates the constructed CKT on the LB($a + 4s, b + 4t, 4$). This completes the proof. $\square$

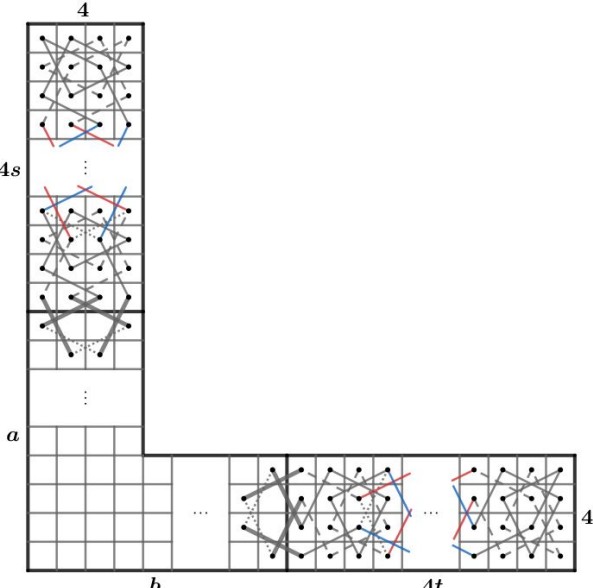

**Figure 8.** A CKT on the $LB(a + 4s, b + 4t, 4)$.

By properly rotating and flipping the $LB(m, n, 4)$, where $m, n \geq 5$, we obtain the following result immediately.

**Corollary 1.** *A $7B(m, n, 4)$ has a CKT containing an edge $(4, 1) - (2, 2)$ for all $m, n \geq 5$.*

We note that Theorem 4 and Corollary 1 will be used in Case 2 of Theorem 8 in Section 4. Next, we construct two OKTs for the $LB(m, n, 3)$ and two OKTs for the $7B(m, n, 3)$ for $m, n \geq 4$.

**Theorem 5.** *Let $m, n \geq 4$.*

(a) *The $LB(m, n, 3)$ contains an OKT from $(1, 2)$ to $(1, 3)$ if and only if (i) $m + n$ is odd and $m + n \geq 11$ or (ii) $m = 5$ and $n = 4$.*

(b) *The $LB(m, n, 3)$ contains an OKT from $(1, 3)$ to $(2, 2)$ if and only if (i) $m + n$ is even and $m + n \geq 12$ or (ii) $m = 6$ and $n = 4$.*

**Proof.** Let $m, n \geq 4$.

(a) We assume that the $LB(m, n, 3)$ contains an OKT from $(1, 2)$ to $(1, 3)$ and let $m + n$ is even; or $m \neq 5$ and $m + n < 11$; or $n \neq 4$ and $m + n < 11$.

If $m + n$ is even, then the numbers of white squares and black squares are not the same. Thus, the two end-points of this OKT must have the same color. However, $(1, 2)$ and $(1, 3)$ are next to each other and have different colors, which is a contradiction.

Let $m \neq 5$ or $n \neq 4$ and $m + n < 11$. By the above argument $m + n$ must be odd and since $m, n \geq 4$, we have $m + n = 9$ which implies that $m = 4$ and $n = 5$.

For $m = 4$ and $n = 5$, let $G$ be a knight graph of the $LB(4, 5, 3)$. Consider $G' = G - \{(1, 2), (1, 3)\}$. Since the $LB(4, 5, 3)$ contains an OKT from $(1, 2)$ to $(1, 3)$, $G'$ has a Hamiltonian path. Let $S = \{(2, 3), (3, 2), (3, 3), (3, 4), (4, 3)\}$. Then, $\omega(G' - S) = 7 > 6 = |S| + 1$ as shown in Figure 9. By Theorem 3, we obtain a contradiction.

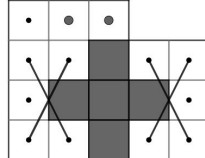

**Figure 9.** Components of $G' - S$.

On the other hand, let us assume that $m + n$ is odd and $m + n \geq 11$; or $m = 5$ and $n = 4$. If $m = 5$ and $n = 4$, then the required OKT presented in Figure 10.

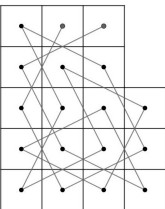

**Figure 10.** Required open knight's tour (OKT) on the LB$(5, 4, 3)$.

If $m + n$ is odd and $m + n \geq 11$, we construct OKTs on some small LB$(m, n, 3)$ according to the remainders of $m$ and $n$ after divided by 4 as the following Figures 11–14.

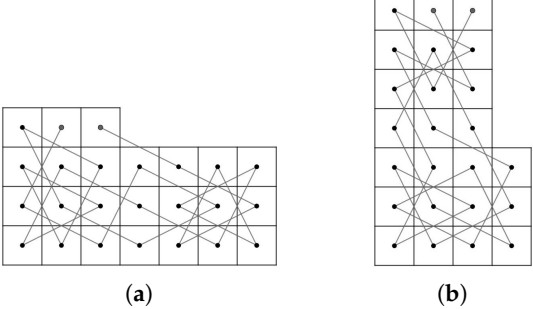

(a)                                    (b)

**Figure 11.** OKTs on the LB$(4, 7, 3)$ and LB$(7, 4, 3)$.

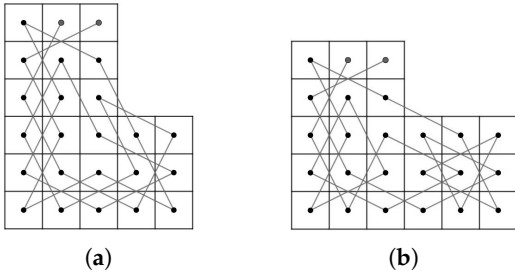

(a)                                    (b)

**Figure 12.** OKTs on the LB$(6, 5, 3)$ and LB$(5, 6, 3)$.

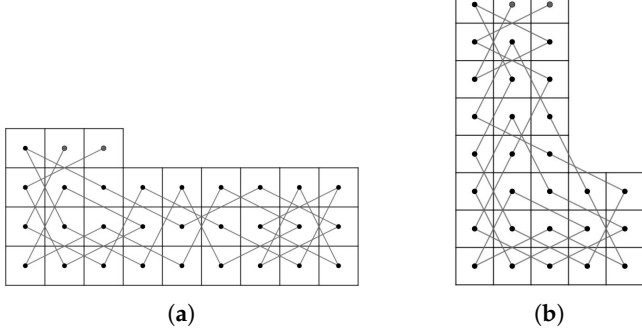

**Figure 13.** OKTs on the LB$(4, 9, 3)$ and LB$(8, 5, 3)$.

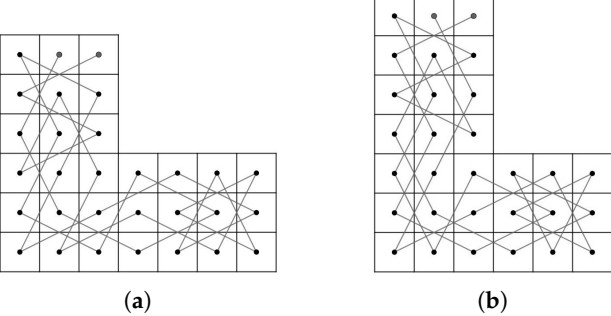

**Figure 14.** OKTs on the LB$(6, 7, 3)$ and LB$(7, 6, 3)$.

Next, for the larger LB, we start by constructing an OKT on CB$(3 \times 4)$ from $(1, 1)$ to $(2, 1)$ that contains an edge $(1, 3) - (3, 4)$ as shown in Figure 15.

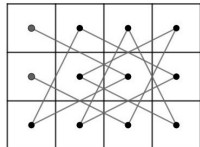

**Figure 15.** An OKT on CB$(3 \times 4)$.

Then, we construct an OKT on the CB$(3 \times 4t)$, where $t \geq 2$. Let us connect $t$ CB$(3 \times 4)$'s in Figure 15 to the right of each other and do the following.

(i)   For $1 \leq i \leq t - 1$, delete $(1, 3) - (3, 4)$ from the OKT of the $i$th CB$(3 \times 4)$;

(ii)  For $1 \leq i \leq t - 1$, join $(1, 3)$ and $(3, 4)$ of the $i$th CB$(3 \times 4)$ to $(2, 1)$ and $(1, 1)$ of the $(i + 1)$th CB$(3 \times 4)$, respectively.

By rotating Figure 16 clockwise for 90 degrees, we also obtain an OKT on CB$(4s \times 3)$ from $(1, 2)$ to $(1, 3)$ as shown in Figures 17.

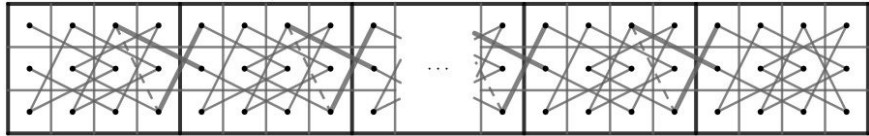

**Figure 16.** An OKT on CB$(3 \times 4t)$.

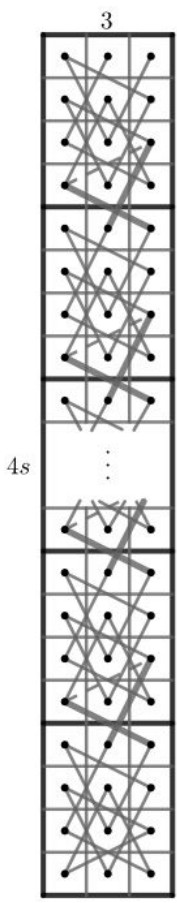

**Figure 17.** An OKT on CB($4s \times 3$).

Now, we are ready to construct an OKT on a larger LB by placing the CB($3 \times 4t$) to the right and the CB($4s \times 3$) above each smaller LB that we have considered before, respectively.

<u>Case 1:</u> there exist nonnegative integers $s, t$ such that $m = 4 + 4s$ and $n = 7 + 4t$. We divided the LB($m, n, 3$) into subboards, CB($4s \times 3$) (Figure 17) and LB($4, 7, 3$) (Figure 11a) if $s > 0$ and $t = 0$ and LB($4, 7, 3$) (Figure 11a) and CB($3 \times 4t$) (Figure 16) if $s = 0$ and $t > 0$. Otherwise, we divide into three subboards, CB($4s \times 3$) (Figure 17), LB($4, 7, 3$) (Figure 11a) and CB($3 \times 4t$) (Figure 16). Then, we construct the required OKT by the following two steps.

(i)　If $s > 0$ and $t = 0$, then delete $(4s, 1) - (4s - 1, 3)$ of the OKT on the CB($4s \times 3$) in Figure 17. If $s = 0$ and $t > 0$, then delete $(2, 6) - (4, 7)$ of the OKT on the LB($4, 7, 3$) in Figure 11a. Otherwise, delete both edges.

(ii)　If $s > 0$ and $t = 0$, then join $(4s, 1)$ and $(4s - 1, 3)$ of the CB($4s \times 3$) to $(1, 3)$ and $(1, 2)$ of the LB($4, 7, 3$), respectively. If $s = 0$ and $t > 0$, then join $(2, 6)$ and $(4, 7)$ of the LB($4, 7, 3$) to $(2, 1)$ and $(1, 1)$ of the CB($3 \times 4t$) chessboard, respectively. Otherwise, join four pairs of vertices together.

<u>Case 2:</u> there exist nonnegative integers $s, t$ such that $m = 7 + 4s$ and $n = 4 + 4t$. Then, we construct the required OKT by using the same procedure as we did in Case 1 but LB($4, 7, 3$), $(2, 6)$ and $(4, 7)$ are replaced by LB($7, 4, 3$) (Figure 11b), $(5, 3)$ and $(7, 4)$, respectively.

<u>Case 3:</u> there exist nonnegative integers $s, t$ such that $m = 6 + 4s$ and $n = 5 + 4t$. Then, we construct the required OKT by using the same procedure as we did in Case 1 but LB($4, 7, 3$), $(2, 6)$ and $(4, 7)$ are replaced by LB($6, 5, 3$) (Figure 12a), $(4, 4)$ and $(6, 5)$, respectively.

<u>Case 4:</u> there exist nonnegative integers $s, t$ such that $m = 5 + 4s$ and $n = 6 + 4t$. Then, we construct the required OKT by using the same procedure as we did in Case 1 but LB($4, 7, 3$), $(2, 6)$ and $(4, 7)$ are replaced by LB($5, 6, 3$) (Figure 12b), $(3, 5)$ and $(5, 6)$, respectively.

<u>Case 5</u>: there exist nonnegative integers $s, t$ such that $m = 4 + 4s$ and $n = 9 + 4t$. Then, we construct the required OKT by using the same procedure as we did in Case 1 but LB(4, 7, 3), (2, 6) and (4, 7) are replaced by LB(4, 9, 3) (Figure 13a), (2, 8) and (4, 9), respectively.

<u>Case 6</u>: there exist nonnegative integers $s, t$ such that $m = 8 + 4s$ and $n = 5 + 4t$. Then, we construct the required OKT by using the same procedure as we did in Case 1 but LB(4, 7, 3), (2, 6) and (4, 7) are replaced by LB(8, 5, 3) (Figure 13b), (6, 4) and (8, 5), respectively.

<u>Case 7</u>: there exist nonnegative integers $s, t$ such that $m = 5 + 4s$ and $n = 4 + 4t$. Then, we construct the required OKT by using the same procedure as we did in Case 1 but LB(4, 7, 3), (2, 6) and (4, 7) are replaced by LB(5, 4, 3) (Figure 10), (3, 3) and (5, 4), respectively.

<u>Case 8</u>: there exist nonnegative integers $s, t$ such that $m = 6 + 4s$ and $n = 7 + 4t$. Then, we construct the required OKT by using the same procedure as we did in Case 1 but LB(4, 7, 3), (2, 6) and (4, 7) are replaced by LB(6, 7, 3) (Figure 14a), (4, 6) and (6, 7), respectively.

<u>Case 9</u>: there exist nonnegative integers $s, t$ such that $m = 7 + 4s$ and $n = 6 + 4t$. Then, we construct the required OKT by using the same procedure as we did in Case 1 but LB(4, 7, 3), (2, 6) and (4, 7) are replaced by LB(7, 6, 3) (Figure 14b), (5, 5) and (7, 6), respectively.

(b) We assume that the LB$(m, n, 3)$ contains an OKT from $(1, 3)$ to $(2, 2)$ and let $m + n$ is odd; or $m \neq 6$ and $m + n < 12$; or $n \neq 4$ and $m + n < 12$.

If $m + n$ is odd, then the numbers of white squares and black squares are the same. Thus, the two end-points of this OKT must have the different color. However, $(1, 3)$ and $(2, 2)$ have the same color, a contradiction. Next, we consider the cases that $m = 4$ and $n = 4$; or $m = 4$ and $n = 6$; or $m = 5$ and $n = 5$.

For $m = 4$ and $n = 4$, let $G_1$ be a knight graph of the LB(4, 4, 3). Consider $G_1' = G_1 - \{(1, 3), (2, 2)\}$. Since the LB(4, 4, 3) contains an OKT from $(1, 3)$ to $(2, 2)$, $G_1'$ has a Hamiltonian path. Let $S = \{(2, 3), (3, 2), (3, 3)\}$. Then, $\omega(G_1' - S) = 5 > 4 = |S| + 1$ as shown in Figure 18. By Theorem 3, we obtain a contradiction.

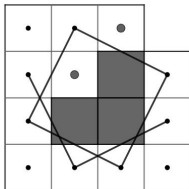

**Figure 18.** Components of $G_1' - S$.

For $m = 4$ and $n = 6$, let $G_2$ be a knight graph of the LB(4, 6, 3). Consider $G_2' = G_2 - \{(1, 3), (2, 2)\}$. Since the LB(4, 6, 3) contains an OKT from $(1, 3)$ to $(2, 2)$, $G_2'$ has a Hamiltonian path. Let $S = \{(2, 3), (2, 4), (3, 3), (3, 4), (4, 3), (4, 4)\}$. Then, $\omega(G_2' - S) = 8 > 7 = |S| + 1$ as shown in Figure 19. By Theorem 3, we obtain a contradiction.

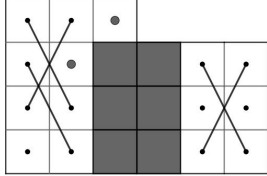

**Figure 19.** Components of $G_2' - S$.

For $m = 5$ and $n = 5$, let $G_3$ be a knight graph of the LB(5, 5, 3). Consider $G_3' = G_3 - \{(1, 3), (2, 2)\}$. Since the LB(5, 5, 3) contains an OKT from $(1, 3)$ to $(2, 2)$, $G_3'$ has a Hamiltonian path. Let $S = \{(2, 3), (3, 2), (3, 3), (4, 2), (4, 3), (5, 3)\}$. Then, $\omega(G_3' - S) = 8 > 7 = |S| + 1$ as shown in Figure 20. By Theorem 3, we obtain a contradiction.

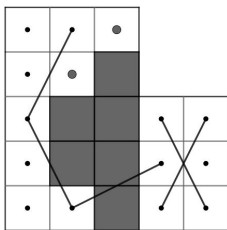

**Figure 20.** Components of $G_3' - S$.

On the other hand, let us assume that $m + n$ is even and $m + n \geq 12$; or $m = 6$ and $n = 4$. If $m = 6$ and $n = 4$, then the required OKT is presented in Figure 21.

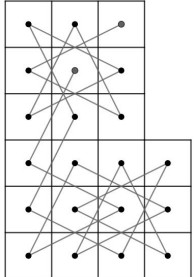

**Figure 21.** Required OKT on the LB$(6, 4, 3)$.

If $m + n$ is even and $m + n \geq 12$, we construct OKTs on some small LB$(m, n, 3)$ according to the remainders of $m$ and $n$ after divided by 4 as the following Figures 22–27.

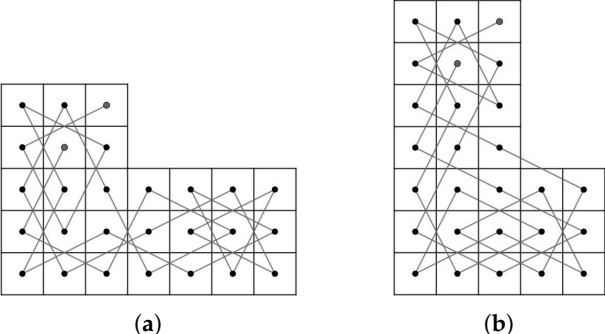

(a)            (b)

**Figure 22.** OKTs on the LB$(5, 7, 3)$ and LB$(7, 5, 3)$.

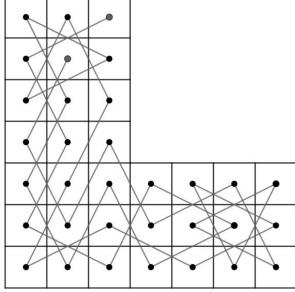

**Figure 23.** An OKT on the LB$(7, 7, 3)$.

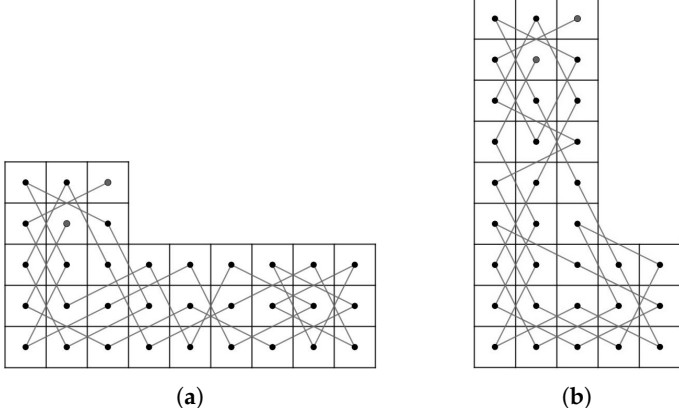

**Figure 24.** OKTs on the LB$(5, 9, 3)$ and LB$(9, 5, 3)$.

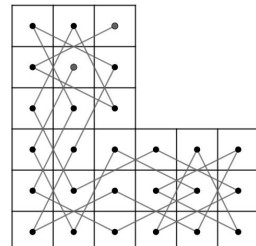

**Figure 25.** An OKT on the LB$(6, 6, 3)$.

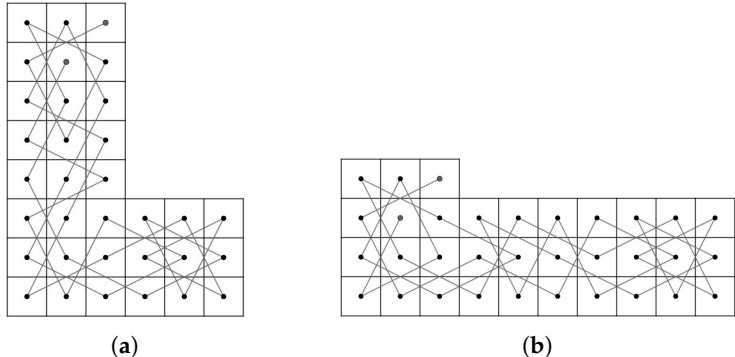

**Figure 26.** OKTs on the LB$(8, 6, 3)$ and LB$(4, 10, 3)$.

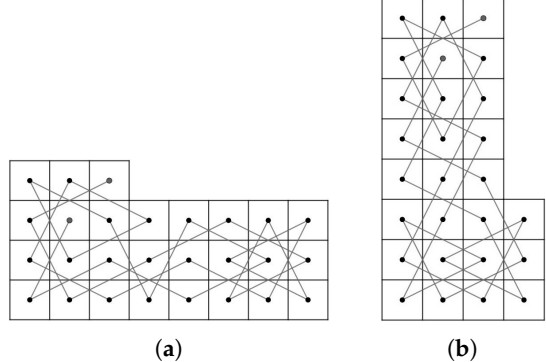

**Figure 27.** OKTs on the LB$(4, 8, 3)$ and CB$(8, 4, 3)$.

Next, for the larger LB, we start by constructing an OKT on CB$(4 \times 3)$ from $(1, 3)$ to $(4, 1)$ and contains an edge $(2, 2) - (4, 3)$ as shown in Figure 28.

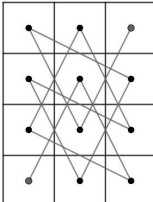

**Figure 28.** An OKT on CB(4 × 3).

Then, we construct two paths on CB(4$s$ × 3), where $s \geq 2$. Let us connect $s$ CB(4 × 3)'s in Figure 28 on the top of each other and do the following.

(i)  For $1 \leq i \leq s$, delete $(2,2) - (4,3)$ from the OKT of the $i$th CB(4 × 3);

(ii)  For $1 \leq i \leq s - 1$, join $(4,1)$ and $(4,3)$ of the $i$th CB(4 × 3) to $(1,3)$ and $(2,2)$ of the $(i+1)$th CB(4 × 3), respectively.

We can see from Figure 29 that either $s$ is odd or $s$ is even, there is one path that has $(1,3)$ as its end-point and another path that has $(2,2)$ as its end-point.

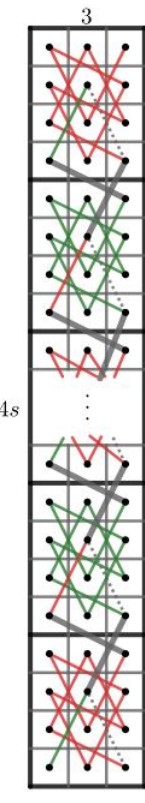

**Figure 29.** Two paths on 4$s$ × 3 chessboard.

Now, we are ready to construct an OKT on a larger LB by placing the CB(3 × 4$t$) in Figure 16 to right and the CB(4$s$ × 3) above each smaller LB that we have considered before, respectively.
<u>Case 1:</u> there exist nonnegative integers $s, t$ such that $m = 5 + 4s$ and $n = 7 + 4t$. We divide the LB($m, n, 3$) into subboards, CB(4$s$ × 3) (Figure 29) and LB(5, 7, 3) (Figure 22a) if $s > 0$ and $t = 0$ and LB(5, 7, 3) (Figure 22a) and CB(3 × 4$t$) (Figure 16) if $s = 0$ and $t > 0$. Otherwise, we divide into three subboards, CB(4$s$ × 3) (Figure 29), LB(5, 7, 3) (Figure 22a) and CB(3 × 4$t$) (Figure 16). Then, we construct the required OKT by the followings.

(i)  If $s \geq 0$ and $t > 0$, then delete $(3,6) - (5,7)$ of the OKT on the LB(5, 7, 3) in Figure 22a.

(ii)　　If $s > 0$ and $t = 0$, then join $(4s, 1)$ and $(4s, 3)$ of the CB($4s \times 3$) in Figure 29 to $(1, 3)$ and $(2, 2)$ of the LB$(5, 7, 3)$, respectively. If $s = 0$ and $t > 0$, then join $(3, 6)$ and $(5, 7)$ of the LB$(5, 7, 3)$ to $(2, 1)$ and $(1, 1)$ of the CB($3 \times 4t$) in Figure 16, respectively. Otherwise, join four pairs of vertices together.

<u>Case 2</u>: there exist nonnegative integers $s, t$ such that $m = 7 + 4s$ and $n = 5 + 4t$. Then, we construct the required OKT by using the same procedure as we did in Case 1 but LB$(5, 7, 3)$, $(3, 6)$ and $(5, 7)$ are replaced by LB$(7, 5, 3)$ (Figure 22b), $(5, 4)$ and $(7, 5)$, respectively.

<u>Case 3</u>: there exist nonnegative integers $s, t$ such that $m = 7 + 4s$ and $n = 7 + 4t$. Then, we construct the required OKT by using the same procedure as we did in Case 1 but LB$(5, 7, 3)$, $(3, 6)$ and $(5, 7)$ are replaced by LB$(7, 7, 3)$ (Figure 23), $(5, 6)$ and $(7, 7)$, respectively.

<u>Case 4</u>: there exist nonnegative integers $s, t$ such that $m = 5 + 4s$ and $n = 9 + 4t$. Then, we construct the required OKT by using the same procedure as we did in Case 1 but LB$(5, 7, 3)$, $(3, 6)$ and $(5, 7)$ are replaced by LB$(5, 9, 3)$ (Figure 24a), $(3, 8)$ and $(5, 9)$, respectively.

<u>Case 5</u>: there exist nonnegative integers $s, t$ such that $m = 9 + 4s$ and $n = 5 + 4t$. Then, we construct the required OKT by using the same procedure as we did in Case 1 but LB$(5, 7, 3)$, $(3, 6)$ and $(5, 7)$ are replaced by LB$(9, 5, 3)$ (Figure 24b), $(7, 4)$ and $(9, 5)$, respectively.

<u>Case 6</u>: there exist nonnegative integers $s, t$ such that $m = 6 + 4s$ and $n = 6 + 4t$. Then, we construct the required OKT by using the same procedure as we did in Case 1 but LB$(5, 7, 3)$, $(3, 6)$ and $(5, 7)$ are replaced by LB$(6, 6, 3)$ (Figure 25), $(4, 5)$ and $(6, 6)$, respectively.

<u>Case 7</u>: there exist nonnegative integers $s, t$ such that $m = 8 + 4s$ and $n = 6 + 4t$. Then, we construct the required OKT by using the same procedure as we did in Case 1 but LB$(5, 7, 3)$, $(3, 6)$ and $(5, 7)$ are replaced by LB$(8, 6, 3)$ (Figure 26a), $(6, 5)$ and $(8, 6)$, respectively.

<u>Case 8</u>: there exist nonnegative integers $s, t$ such that $m = 4 + 4s$ and $n = 10 + 4t$. Then, we construct the required OKT by using the same procedure as we did in Case 1 but LB$(5, 7, 3)$, $(3, 6)$ and $(5, 7)$ are replaced by LB$(4, 10, 3)$ (Figure 26b), $(2, 9)$ and $(4, 10)$, respectively.

<u>Case 9</u>: there exist nonnegative integers $s, t$ such that $m = 6 + 4s$ and $n = 4 + 4t$. Then, we construct the required OKT by using the same procedure as we did in Case 1 but LB$(5, 7, 3)$, $(3, 6)$ and $(5, 7)$ are replaced by LB$(6, 4, 3)$ (Figure 21), $(4, 3)$ and $(6, 4)$, respectively.

<u>Case 10</u>: there exist nonnegative integers $s, t$ such that $m = 4 + 4s$ and $n = 8 + 4t$. Then, we construct the required OKT by using the same procedure as we did in Case 1 but LB$(5, 7, 3)$, $(3, 6)$ and $(5, 7)$ are replaced by LB$(4, 8, 3)$ (Figure 27a), $(2, 7)$ and $(4, 8)$, respectively.

<u>Case 11</u>: there exist nonnegative integers $s, t$ such that $m = 8 + 4s$ and $n = 4 + 4t$. Then, we construct the required OKT by using the same procedure as we did in Case 1 but LB$(5, 7, 3)$, $(3, 6)$ and $(5, 7)$ are replaced by LB$(8, 4, 3)$ (Figure 27b), $(6, 3)$ and $(8, 4)$, respectively.

　　This completes the proof.　□

　　Next, we get the following Corollary by flipping and rotating 90 degrees clockwise the LB in the above Theorem.

**Corollary 2.** *Let $m, n \geq 4$.*

(a)　　*The 7B$(m, n, 3)$ contains an OKT from $(2, 1)$ to $(3, 1)$ if and only if (i) $m + n$ is odd and $m + n \geq 11$ or (ii) $m = 4$ and $n = 5$.*

(b)　　*The 7B$(m, n, 3)$ contains an OKT from $(3, 1)$ to $(2, 2)$ if and only if (i) $m + n$ is even and $m + n \geq 12$ or (ii) $m = 4$ and $n = 6$.*

　　We note that Theorem 5(b) and Corollary 2(b) will be used in Case 1.1 of Theorem 8 in Section 4. While, Theorem 5(a) and Corollary 2(a) will be used in Case 1.2 of Theorem 8 in Section 4.

### 3. Existence of a Special OKT on CB($m \times n$)

The following theorem gives necessary and sufficient conditions on the existence of a special OKT on CB($m \times n$) from $(m, 1)$ to $(2, n - 1)$. This OKT will be used to prove our main result for $r \geq 5$ when $r$ is odd (Case 3.2 of Theorem 8 in Section 4).

**Theorem 6.**

(a) Let $m \leq 4$ and $n \geq m$. Then, a CB($m \times n$) contains an OKT from $(m, 1)$ to $(2, n - 1)$ if and only if $m = 3$ and $n \geq 7$.

(b) Let $n \geq m \geq 5$. Then, a CB($m \times n$) contains an OKT from $(m, 1)$ to $(2, n - 1)$ if and only if $m$ and $n$ are not both even.

**Proof.** (a) Let $m \leq 4$. We assume that a CB($m \times n$) contains an OKT from $(m, 1)$ to $(2, n - 1)$ and let $m \neq 3$; or $n \leq 6$. Then, we consider 4 cases as follows.

**Case 1:** $m = 1$ and $n \geq 1$ or $m = 2$ and $n \geq 2$ or $m = 3$ and $n \in \{3, 5, 6\}$. A CB($m \times n$) contains no OKT by using Theorem 2, contradiction.

**Case 2:** For $m = 3$ and $n = 4$, let $G_1$ be a knight graph of the CB($3 \times 4$). We assume that $G_1$ contains a Hamiltonain path from $(3, 1)$ to $(2, 3)$. Consider $G_1' = G_1 - \{(2, 3)\}$. By assumption, $G_1'$ has a Hamiltonian path. Let $S = \{(1, 2), (3, 2)\}$. Then, $\omega(G_1' - S) = 4 > 3 = |S| + 1$ as shown in Figure 30. By Theorem 3, we obtain a contradiction.

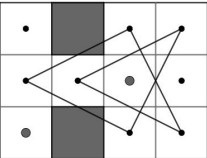

**Figure 30.** Components of $G_1' - S$.

**Case 3:** For $m = 4$ and $n$ is odd such that $n \geq 5$. Let $G_2$ be a knight graph of the CB($4 \times n$). We assume that $G_2$ contains a Hamiltonian path from $(m, 1)$ to $(2, n - 1)$. Consider $G_2' = G_2 - \{(2, n - 1)\}$. Let $S = \{(2, j), (3, l) \mid j$ is even, $2 \leq j \leq n - 3, l$ is odd and $1 \leq l \leq n\}$. Then, we can use mathematical induction to show that $\omega(G_2' - S) = n + 1 > n = |S| + 1$ as shown in Figure 31. By Theorem 3, we have a contradiction.

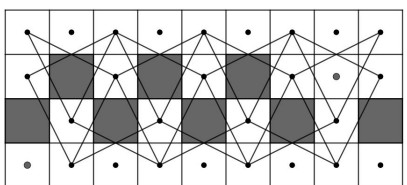

**Figure 31.** Components of $G_2' - S$, where $n = 9$.

**Case 4:** For $m = 4$ and $n$ is even such that $n \geq 4$. Assume that CB($4 \times n$) contains an OKT from $(4, 1)$ to $(2, n - 1)$. Since CB($4 \times n$) contains the same numbers of black and white squares, this OKT must have end-points at two squares with different color. However, $4 + 1 = 5$ and $2 + n - 1 = n + 1$ are odd. Thus, $(m, 1)$ and $(2, n - 1)$ are two squares of the same color, contradiction.

On the other hand, let us assume that $m = 3$ and $n \geq 7$.

Let us construct OKTs from $(3, 1)$ to $(2, n - 1)$ on some small size CB($m \times n$) where $n \in \{7, 8, 9, 10\}$ as shown in Figure 32.

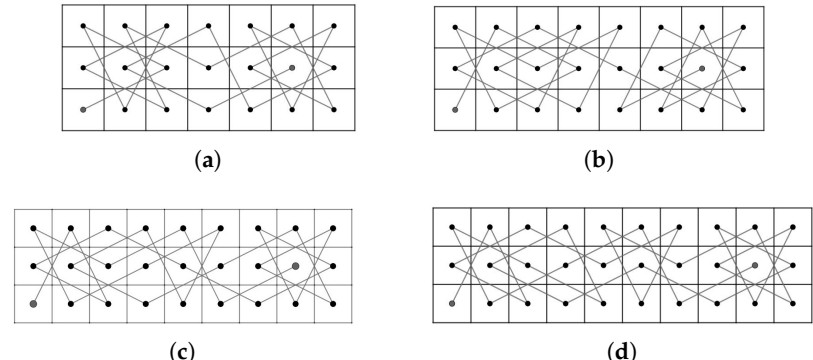

**Figure 32.** OKTs from $(3,1)$ to $(2, n-1)$ on the $CB(m \times n)$ where $n \in \{7, 8, 9, 10\}$.

Before we continue further, let us rotate the $CB(4 \times 3)$ shown in Figure 28 for 90 degrees clockwise and flip it to obtain an OKT from $(3,1)$ to $(1,4)$ on $CB(3 \times 4)$. We can place $t$ of these $CB(3 \times 4)$ to the right of each other to extend this OKT into an OKT on $CB(3 \times 4t)$ by connecting $(1,4)$ on the $i$th $CB(3 \times 4)$ to $(3,1)$ on the $(i+1)$th $CB(3 \times 4)$ for all $1 \le i \le t-1$ as shown in Figure 33. Note that this extended OKT starts from $(3,1)$ to $(1,4t)$.

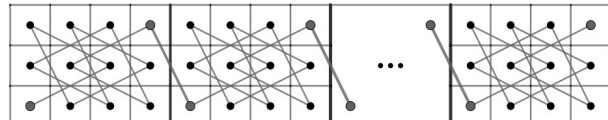

**Figure 33.** An OKT from $(3,1)$ to $(1,4t)$ on $CB(3 \times 4t)$.

Next, let $n$ be a positive integer such that $n \ge 11$.

If $n \equiv 3 \pmod 4$ (respectively, $n \equiv 0 \pmod 4$, $n \equiv 1 \pmod 4$, $n \equiv 2 \pmod 4$)), then there is a positive integer $t$ such that $n = 7 + 4t$ (respectively, $n = 8 + 4t, n = 9 + 4t, n = 10 + 4t$). We divide the $CB(3 \times n)$ into subboards, $CB(3 \times 4t)$ (Figure 33) and $CB(3 \times 7)$ (Figure 32a) (respectively, $CB(3 \times 8)$ (Figure 32b), $CB(3 \times 9)$ (Figure 32c), $CB(3 \times 10)$ (Figure 32d)). Then, we construct the required OKT by connecting $(1, 4t)$ of the OKT on the $CB(3 \times 4t)$ in Figure 33 to $(3,1)$ of the OKT on the $CB(3 \times 7)$ in Figure 32a (respectively, $CB(3 \times 8)$ in Figure 32b, $CB(3 \times 9)$ in Figure 32c, $CB(3 \times 10)$ in Figure 32d).

(b) Let $n \ge m \ge 5$. We assume that a $CB(m \times n)$ contains an OKT from $(m, 1)$ to $(2, n-1)$ and let $m$ and $n$ are both even. Since $CB(m \times n)$ contains the same numbers of black and white squares, this OKT must have end-points at two squares with different colors. However, $m + 1$ and $2 + n - 1 = n + 1$ are odd. Thus, $(m, 1)$ and $(2, n-1)$ are two squares of the same color, contradiction.

On the other hand, let us assume that $m$ and $n$ are not both even such that $n \ge m \ge 5$. Then, we consider three cases as follows.

**Case 1:** $m$ and $n$ are both odd such that $m, n \ge 5$. Let us construct OKTs from $(m, 1)$ to $(2, n-1)$ containing the edge $(1, n) - (3, n-1)$ on some small size $CB(m \times n)$ where $m, n \in \{5, 7\}$ as shown in Figure 34.

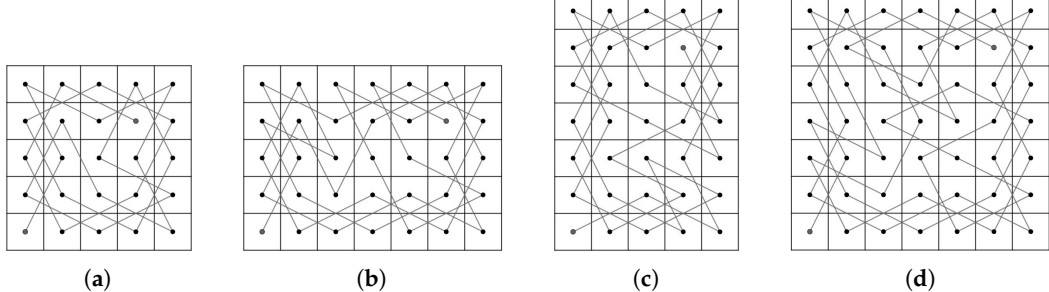

**Figure 34.** OKTs from $(m, 1)$ to $(2, n-1)$ on the CB$(m \times n)$ where $m, n \in \{5, 7\}$.

For the larger CB$(m \times n)$, we start by constructing two paths on CB$(m \times 4)$. The first path starts from $(1, 1)$ to $(2, 3)$ and the second path starts from $(2, 2)$ to $(4, 1)$ where $n \in \{5, 6, 7, 8\}$ as shown in Figure 35.

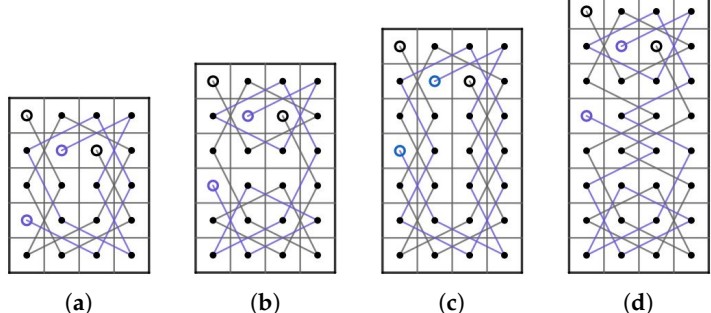

**Figure 35.** Two paths on CB$(m \times 4)$ where $m \in \{5, 6, 7, 8\}$.

Next, we construct an OKT from $(1, 3)$ to $(4, 1)$ containing the edges $(1, n) - (3, n-1)$ and $(2, n-1) - (4, n)$ on the CB$(4 \times n)$ where $n \in \{5, 6, 7, 8\}$ as shown in Figure 36.

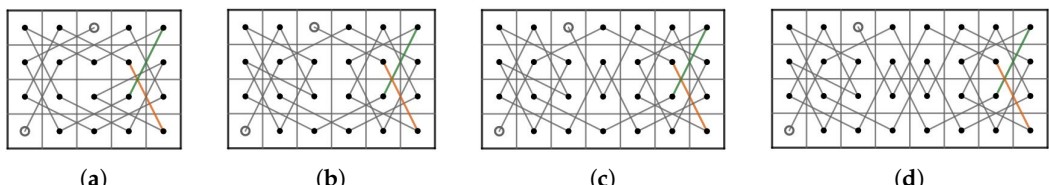

**Figure 36.** OKTs on CB$(4 \times n)$ where $n \in \{5, 6, 7, 8\}$.

Let $m, n$ be odd integers such that $m \geq 5$ and $n \geq 9$.

**Case 1.1:** $m, n \equiv 1 \pmod{4}$. There are integers $s$ and $t$ such that $0 \leq s \leq t$, $t \neq 0$, $m = 5 + 4s$ and $n = 5 + 4t$. If $s = 0$, then we divide the CB$(5 \times n)$ into subboards, CB$(5 \times 5)$ (Figure 34a) and $t$ CB$(5 \times 4)$'s (Figure 35a). Then, we construct the required OKT by the followings.

(i) We delete $(1, 5) - (3, 4)$ of the OKT on the CB$(5 \times 5)$ and connect $(2, 4), (1, 5)$ and $(3, 4)$ of the CB$(5 \times 5)$ to $(1, 1), (2, 2)$ and $(4, 1)$ of the 1st CB$(5 \times 4)$, respectively.

(ii) We delete $(1, 4) - (3, 3)$ of the second path of the $i$th CB$(5 \times 4)$ for all $1 \leq i \leq t - 1$. Then, we connect $(2, 3), (1, 4)$ and $(3, 3)$ of the $i$th CB$(5 \times 4)$ to $(1, 1), (2, 2)$ and $(4, 1)$ of the $(i + 1)$th CB$(5 \times 4)$.

If $s > 0$, then we divide the CB$(m \times n)$ into subboards, CB$(5 \times n)$ (Figure 37) and $s$ CB$(4 \times n)$'s from the top to the bottom. Then, we construct the required OKT by the followings.

(i') For each $1 \leq i \leq s$, we divide the *i*th CB($4 \times n$) into subboards, CB($4 \times 5$) (Figure 36a) and CB($4 \times 4t$) (Figure 6). Delete $(1,5) - (3,4)$ and $(2,4) - (4,5)$ of the OKT on CB($4 \times 5$). Then, join $(2,4), (4,5), (1,5)$ and $(3,4)$ of the CB($4 \times 5$) to $(1,1), (2,1), (3,1)$ and $(4,1)$ of the CB($4 \times 4t$), respectively, to obtain an OKT on the CB($4 \times n$) as shown in Figure 38.

(ii') Join $(5,1)$ of the OKT on CB($5 \times n$) to $(1,3)$ of the 1st CB($4 \times n$) shown in Figure 38.

(iii') For each $1 \leq i \leq s-1$, we join $(4,1)$ of the OKT on the *i*th CB($4 \times n$) (Figure 38) to $(1,3)$ of the OKT on the $(i+1)$th CB($4 \times n$) (Figure 38).

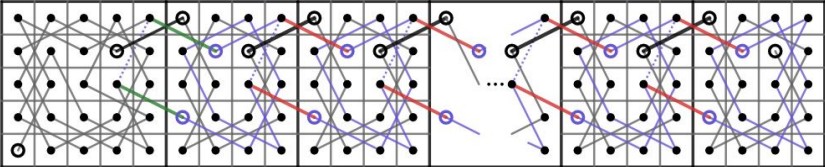

**Figure 37.** An OKT on CB($5 \times n$) in Case 1.1.

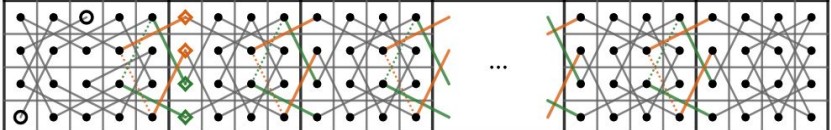

**Figure 38.** An OKT on CB($4 \times n$) in Case 1.1.

**Case 1.2:** $m \equiv 1 \pmod 4$ and $n \equiv 3 \pmod 4$. There are integers $s$ and $t$ such that $0 \leq s \leq t$, $t \neq 0$, $m = 5 + 4s$ and $n = 7 + 4t$. If $s = 0$, then we divide the CB($5 \times n$) into subboards, CB($5 \times 7$) (Figure 34b) and $t$ CB($5 \times 4$)'s (Figure 35a). Then, we construct the required OKT by (i) and (ii) in Case 1.1 but replace CB($5 \times 5$) by CB($5 \times 7$) (Figure 34b) and $(1,5), (3,4)$ and $(2,4)$ by $(1,7), (3,6)$ and $(2,6)$, respectively.

If $s > 0$, then we divide the CB($m \times n$) into subboards, CB($5 \times n$) (Figure 39) and $s$ CB($4 \times n$)'s (Figure 40) from the top to the bottom. Then, we construct the required OKT by (i'), (ii') and (iii') in Case 1.1 but in (i') replace CB($4 \times 5$) by CB($4 \times 7$) (Figure 36c) and $(1,5), (3,4), (2,4)$ and $(4,5)$ by $(1,7), (3,6), (2,6)$ and $(4,7)$, respectively.

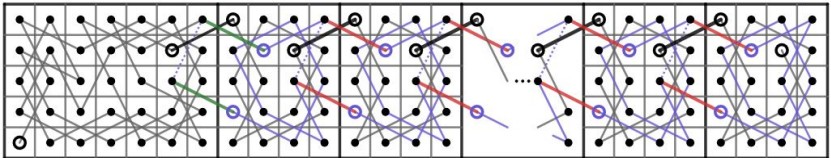

**Figure 39.** An OKT on CB($5 \times n$) in Case 1.2.

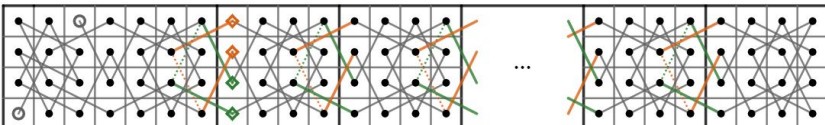

**Figure 40.** An OKT on CB($4 \times n$) in Case 1.2.

**Case 1.3:** $m \equiv 3 \pmod 4$ and $n \equiv 1 \pmod 4$. There are integers $s$ and $t$ such that $0 \leq s < t$, $m = 7 + 4s$ and $n = 5 + 4t$. If $s = 0$, then we divide the CB($7 \times n$) into subboards, CB($7 \times 5$) (Figure 34c) and $t$ CB($7 \times 4$)'s (Figure 35c). Then, we construct the required OKT by (i) and (ii) in Case 1.1 but replace CB($5 \times 5$) and CB($5 \times 4$) by CB($7 \times 5$) (Figure 34c) and CB($7 \times 4$) (Figure 35c), respectively.

If $s > 0$, then we divide the CB($m \times n$) into subboards, CB($7 \times n$) (Figure 41) and $s$ CB($4 \times n$) (Figure 38) from the top to the bottom. Then, we construct the required OKT by (i'), (ii') and (iii') in Case 1.1 but in (ii') replace CB($5 \times n$) by CB($7 \times n$) (Figure 41) and $(5,1)$ by $(7,1)$.

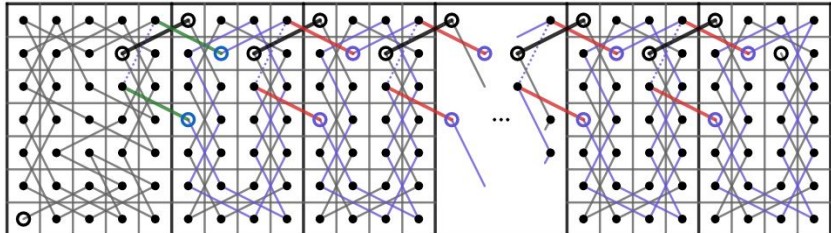

**Figure 41.** An OKT on CB($7 \times n$) in Case 1.3.

**Case 1.4:** $m \equiv 3$ (mod 4) and $n \equiv 3$ (mod 4). There are integers $s$ and $t$ such that $0 \leq s \leq t$, $t \neq 0$, $m = 7 + 4s$ and $n = 7 + 4t$. If $s = 0$, then we divide the CB($7 \times n$) into subboards, CB($7 \times 7$) (Figure 34d) and $t$ CB($7 \times 4$)'s (Figure 35c). Then, we construct the required OKT by (i) and (ii) in Case 1.1 but replace CB($5 \times 5$) and CB($5 \times 4$) by CB($7 \times 7$) (Figure 34d) and CB($7 \times 4$) (Figure 35c)) and $(1, 5)$, $(3, 4)$ and $(2, 4)$ by $(1, 7)$, $(3, 6)$ and $(2, 6)$, respectively.

If $s > 0$, then we divide the CB($m \times n$) into subboards, CB($7 \times n$) (Figure 42) and $s$ CB($4 \times n$)'s (Figure 40) from the top to the bottom. Then, we construct the required OKT by (i'), (ii') and (iii') in Case 1.1 but in (i') and (ii') replace CB($4 \times 5$) and CB($5 \times n$) by CB($4 \times 7$) (Figure 36c) and CB($7 \times n$) (Figure 42) and replace $(1, 5)$, $(3, 4)$, $(2, 4)$, $(4, 5)$ and $(5, 1)$ by $(1, 7)$, $(3, 6)$, $(2, 6)$, $(4, 7)$ and $(7, 1)$, respectively.

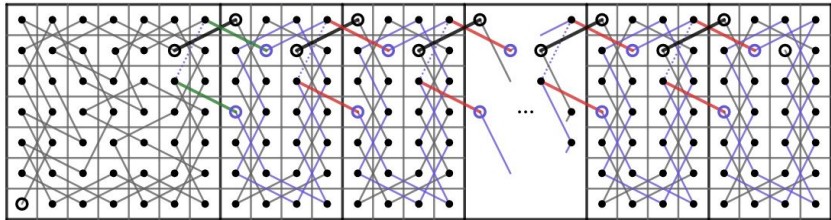

**Figure 42.** An OKT on CB($7 \times n$) in Case 1.4.

**Case 2:** $m$ is odd such that $m \geq 5$ and $n$ is even such that $n \geq 6$. Let us construct OKTs from $(m, 1)$ to $(2, n - 1)$ containing the edge $(1, n) - (3, n - 1)$ on some small size CB($m \times n$) where $m \in \{5, 7\}$ and $n \in \{6, 8\}$ as shown in Figure 43.

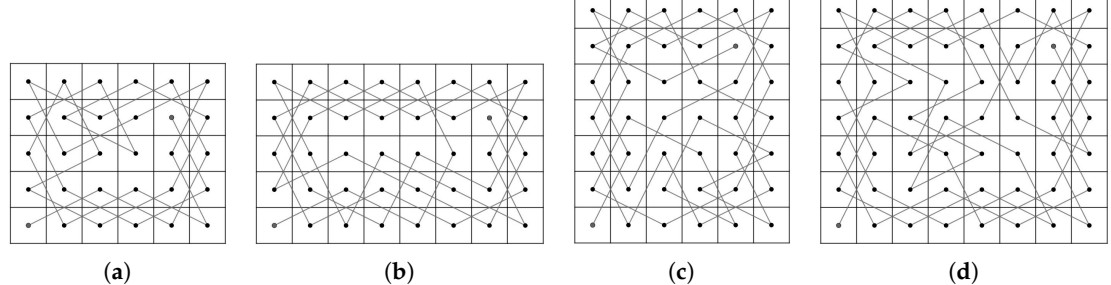

| (a) | (b) | (c) | (d) |

**Figure 43.** OKTs from $(m, 1)$ to $(2, n - 1)$ on the CB($m \times n$) where $m \in \{5, 7\}$ and $n \in \{6, 8\}$.

Let $m$ be an odd integer such that $m \geq 5$ and let $n$ be an even integer such that $n \geq 10$.

**Case 2.1:** $m \equiv 1$ (mod 4) and $n \equiv 0$ (mod 4). There are integers $s$ and $t$ such that $0 \leq s \leq t$, $t \neq 0$, $m = 5 + 4s$ and $n = 8 + 4t$. If $s = 0$, then we divide the CB($5 \times n$) into subboards, CB($5 \times 8$) (Figure 43b) and $t$ CB($5 \times 4$)'s (Figure 35a). Then, we construct the required OKT by (i) and (ii) in Case 1.1 but replace CB($5 \times 5$) by CB($5 \times 8$) (Figure 43b) and $(1, 5)$, $(3, 4)$ and $(2, 4)$ by $(1, 8)$, $(3, 7)$ and $(2, 7)$, respectively.

If $s > 0$, then we divide the CB($m \times n$) into subboards, CB($5 \times n$) (Figure 44) and $s$ CB($4 \times n$)'s (Figure 45) from the top to the bottom. Then, we construct the required OKT by (i'), (ii') and (iii') in

Case 1.1 but in (i′) replace CB(4 × 5) by CB(4 × 8) (Figure 36d) and (1, 5), (3, 4), (2, 4) and (4, 5) by (1, 8), (3, 7), (2, 7) and (4, 8), respectively.

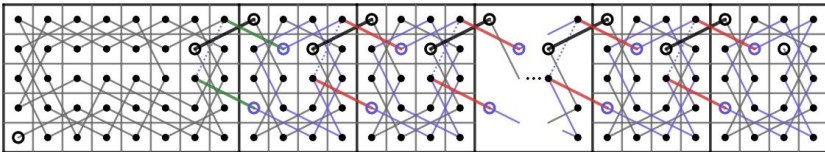

**Figure 44.** An OKT on CB(5 × *n*) in Case 2.1.

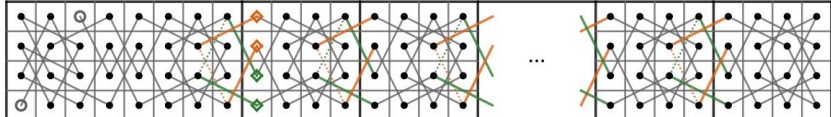

**Figure 45.** An OKT on CB(4 × *n*) in Case 2.1.

**Case 2.2:** *m* ≡ 1 (mod 4) and *n* ≡ 2 (mod 4). There are integers *s* and *t* such that 0 ≤ *s* ≤ *t*, *t* ≠ 0, *m* = 5 + 4*s* and *n* = 6 + 4*t*. If *s* = 0, then we divide the CB(5 × *n*) into subboards, CB(5 × 6) (Figure 43a) and *t* CB(5 × 4)'s (Figure 35a). Then, we construct the required OKT by (i) and (ii) in Case 1.1 but replace CB(5 × 5) by CB(5 × 6) (Figure 43a) and (1, 5), (3, 4) and (2, 4) by (1, 6), (3, 5) and (2, 5), respectively.

If *s* > 0, then we divide the CB(*m* × *n*) into subboards, CB(5 × *n*) (Figure 46) and *s* CB(4 × *n*)'s (Figure 47) from the top to the bottom. Then, we construct the required OKT by (i′), (ii′) and (iii′) in Case 1.1 but in (i′) replace CB(4 × 5) by CB(4 × 6) (Figure 36b) and (1, 5), (3, 4), (2, 4) and (4, 5) by (1, 6), (3, 5), (2, 5) and (4, 6), respectively.

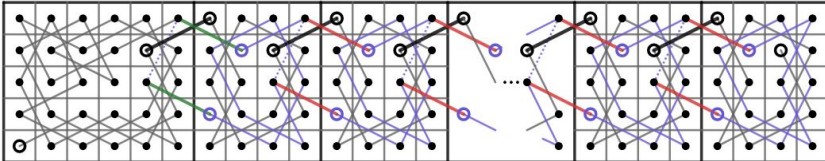

**Figure 46.** An OKT on CB(5 × *n*) in Case 2.2.

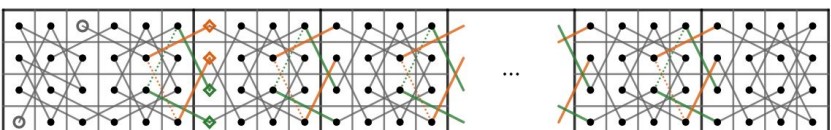

**Figure 47.** An OKT on CB(4 × *n*) in Case 2.2.

**Case 2.3:** *m* ≡ 3 (mod 4) and *n* ≡ 0 (mod 4). There are integers *s* and *t* such that 0 ≤ *s* ≤ *t*, *t* ≠ 0, *m* = 7 + 4*s* and *n* = 8 + 4*t*. If *s* = 0, then we divide CB(7 × *n*) into subboards, CB(7 × 8) (Figure 43d) and *t* CB(7 × 4)'s (Figure 35c). Then, we construct the required OKT by (i) and (ii) in Case 1.1 but replace CB(5 × 5) and CB(5 × 4) by CB(7 × 8) (Figure 43d) and CB(7 × 4) (Figure 35c) and (1, 5), (3, 4) and (2, 4) by (1, 8), (3, 7) and (2, 7), respectively.

If *s* > 0, then we divide the CB(*m* × *n*) into subboards, CB(7 × *n*) (Figure 48) and *s* CB(4 × *n*)'s (Figure 45) from the top to the bottom. Then, we construct the required OKT by (i′), (ii′) and (iii′) in Case 1.1 but in (i′) and (ii′) replace CB(4 × 5) and CB(5 × *n*) by CB(4 × 8) (Figure 36d) and CB(7 × *n*) (Figure 48) and (1, 5), (3, 4), (2, 4) and (4, 5) by (1, 8), (3, 7), (2, 7) and (4, 8), respectively.

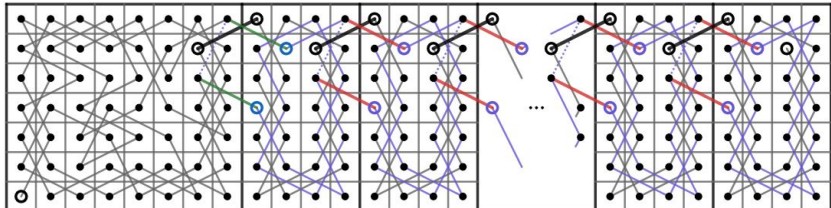

**Figure 48.** An OKT on CB($7 \times n$.) in Case 2.3.

**Case 2.4:** $m \equiv 3 \pmod 4$ and $n \equiv 2 \pmod 4$. There are integers $s$ and $t$ such that $0 \leq s < t$, $m = 7 + 4s$ and $n = 6 + 4t$. If $s = 0$, then we divide the CB($7 \times n$) into subboards, CB($7 \times 6$) (Figure 43c) and $t$ CB($7 \times 4$)'s (Figure 35c). Then, we construct the required OKT by (i) and (ii) in Case 1.1. but replace CB($5 \times 5$) and CB($5 \times 4$) by CB($7 \times 6$) (Figure 43c) and CB($7 \times 4$) (Figure 35c) and $(1, 5)$, $(3, 4)$ and $(2, 4)$ by $(1, 6)$, $(3, 5)$ and $(2, 5)$, respectively.

If $s > 0$, then we divide the CB($m \times n$) into subboards, CB($7 \times n$) (Figure 49) and $s$ CB($4 \times n$)'s (Figure 47) from the top to the bottom. Then, we construct the required OKT by (i'), (ii') and (iii') in Case 1.1. but in (i') and (ii') replace CB($4 \times 5$) and CB($5 \times n$) by CB($4 \times 6$) (Figure 36b) and CB($7 \times n$) (Figure 49) and $(1, 5)$, $(3, 4)$, $(2, 4)$, $(4, 5)$ and $(5, 1)$ by $(1, 6)$, $(3, 5)$, $(2, 5)$, $(4, 6)$ and $(7, 1)$, respectively.

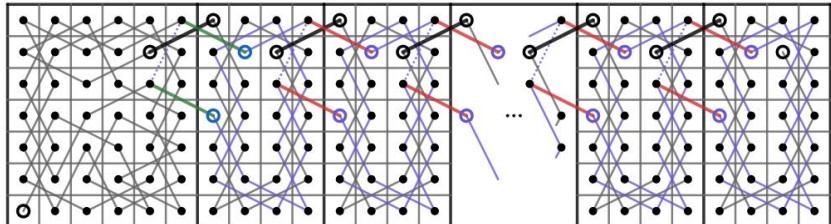

**Figure 49.** An OKTs on CB($7 \times n$) in Case 2.4.

**Case 3:** $m$ is even such that $m \geq 6$ and $n$ is odd such that $n \geq 5$. Let us construct OKTs from $(m, 1)$ to $(2, n - 1)$ containing the edge $(1, n) - (3, n - 1)$ on some small size CB($m \times n$) where $m \in \{6, 8\}$ and $n \in \{5, 7\}$ as shown in Figure 50.

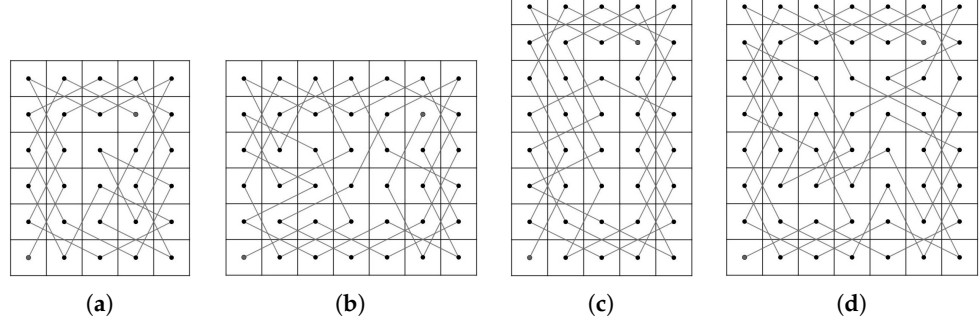

|     |     |     |     |
| :-: | :-: | :-: | :-: |
| (a) | (b) | (c) | (d) |

**Figure 50.** OKTs from $(m, 1)$ to $(2, n - 1)$ on the CB($m \times n$) where $m \in \{6, 8\}$ and $n \in \{5, 7\}$.

Let $m$ be an even integer such that $m \geq 6$ and let $n$ be an odd integer such that $n \geq 9$.

**Case 3.1:** $m \equiv 0 \pmod 4$ and $n \equiv 1 \pmod 4$. There are integers $s$ and $t$ such that $0 \leq s < t$, $m = 8 + 4s$ and $n = 5 + 4t$. If $s = 0$, then we divide the CB($8 \times n$) into subboards, CB($8 \times 5$) (Figure 50c) and $t$ CB($8 \times 4$) (Figure 35d). Then, we construct the required OKT by (i) and (ii) in Case 1.1 replace CB($5 \times 5$) and CB($5 \times 4$) by CB($8 \times 5$) (Figure 50c) and CB($8 \times 4$) (Figure 35d), respectively.

If $s > 0$, then we divide the CB($m \times n$) into subboards, CB($8 \times n$) (Figure 51) and $s$ CB($4 \times n$)'s (Figure 38) from the top to the bottom. Then, we construct the required OKT by (i'), (ii') and (iii') in Case 1.1 but in (ii') replace CB($5 \times n$) by CB($8 \times n$) (Figure 51) and $(5, 1)$ by $(8, 1)$.

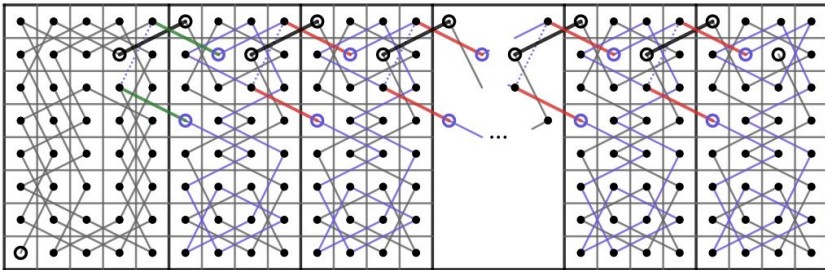

**Figure 51.** An OKTs on CB($8 \times n$) in Case 3.1.

**Case 3.2:** $m \equiv 0 \pmod 4$ and $n \equiv 3 \pmod 4$. There are integers $s$ and $t$ such that $0 \le s < t$, $t \ne 0$, $m = 8 + 4s$ and $n = 7 + 4t$. If $s = 0$, then we divide the CB($8 \times n$) into subboards, CB($8 \times 7$) (Figure 50d) and $t$ CB($8 \times 4$)'s (Figure 35d). Then, we construct the required OKT by (i) and (ii) in Case 1.1 but replace CB($5 \times 5$) and CB($5 \times 4$) by CB($8 \times 7$) (Figure 50d) and CB($8 \times 4$) (Figure 35d) and $(1, 5)$, $(3, 4)$ and $(2, 4)$ by $(1, 7)$, $(3, 6)$ and $(2, 6)$, respectively.

If $s > 0$, then we divide the CB($m \times n$) into subboards, CB($8 \times n$) (Figure 52) and $s$ CB($4 \times n$)'s (Figure 40) from the top to the bottom. Then, we construct the required OKT by (i'), (ii') and (iii') in Case 1.1 but in (i') and (ii') replace CB($4 \times 5$) and CB($5 \times n$) by CB($4 \times 7$) (Figure 36c) and CB($8 \times n$) (Figure 52) and $(1, 5)$, $(3, 4)$, $(2, 4)$, $(4, 5)$ and $(5, 1)$ by $(1, 7)$, $(3, 6)$, $(2, 6)$, $(4, 7)$ and $(8, 1)$, respectively.

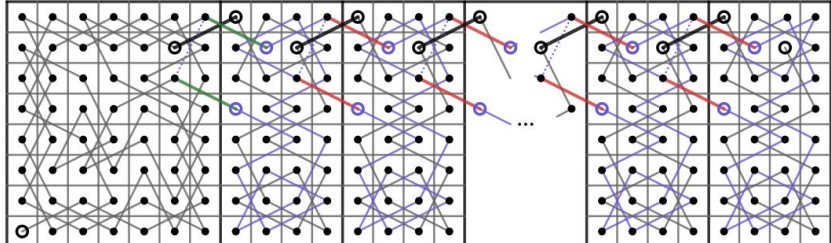

**Figure 52.** An OKT on CB($8 \times n$) in Case 3.2.

**Case 3.3:** $m \equiv 2 \pmod 4$ and $n \equiv 1 \pmod 4$. There are integers $s$ and $t$ such that $0 \le s < t$, $t \ne 0$, $m = 6 + 4s$ and $n = 5 + 4t$. If $s = 0$, then we divide the CB($6 \times n$) into subboards, CB($6 \times 5$) (Figure 50a) and $t$ CB($6 \times 4$) (Figure 35b). Then, we construct the required OKT by (i) and (ii) in Case 1.1 but replace CB($5 \times 5$) and CB($5 \times 4$) by CB($6 \times 5$) (Figure 50a) and CB($6 \times 4$) (Figure 35b), respectively.

If $s > 0$, then we divide the CB($m \times n$) into subboards, CB($6 \times n$) (Figure 53) and $s$ CB($4 \times n$)'s (Figure 38) from the top to the bottom. Then, we construct the required OKT by (i'), (ii') and (iii') in Case 1.1 but in (ii') replace CB($5 \times n$) by CB($6 \times n$) (Figure 53) and $(5, 1)$ by $(6, 1)$.

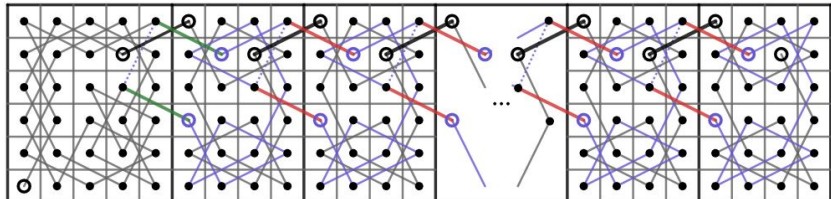

**Figure 53.** An OKT on CB($6 \times n$) in Case 3.3.

**Case 3.4:** $m \equiv 2 \pmod 4$ and $n \equiv 3 \pmod 4$. There are integers $s$ and $t$ such that $0 \le s \le t$, $t \ne 0$, $m = 6 + 4s$ and $n = 7 + 4t$. If $s = 0$, then we divide the CB($6 \times n$) into subboards, CB($6 \times 7$) (Figure 50b) and $t$ CB($6 \times 4$) (Figure 35b). Then, we construct the required OKT by (i) and (ii) in Case 1.1 but replace CB($5 \times 5$) and CB($5 \times 4$) by CB($6 \times 7$) (Figure 50b) and CB($6 \times 4$) (Figure 35b) and $(1, 5)$, $(3, 4)$ and $(2, 4)$ by $(1, 7)$, $(3, 6)$ and $(2, 6)$, respectively.

If $s > 0$, then we divide the CB($m \times n$) into subboards, CB($6 \times n$) (Figure 54) and $s$ CB($4 \times n$)'s (Figure 40) from the top to the bottom. Then, we construct the required OKT by (i'), (ii') and (iii') in

Case 1.1 but in (i′) and (ii′) replace CB(4 × 5) and CB(5 × n) by CB(4 × 7) (Figure 36c) and CB(6 × n) (Figure 54) and (1, 5), (3, 4), (2, 4), (4, 5) and (5, 1) by (1, 7), (3, 6), (2, 6), (4, 7) and (6, 1), respectively.

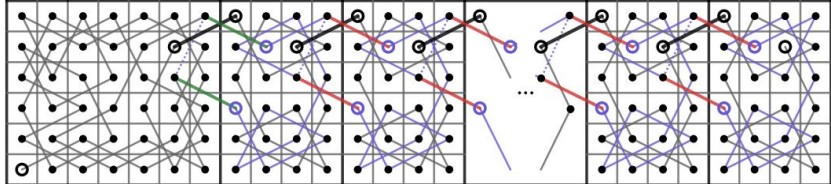

**Figure 54.** An OKT on CB(6 × n) in Case 3.4.

This completes the proof. □

## 4. Main Theorem

To characterize the RB(m, n, r) according to the existence of its CKT, let us first consider the case when r = 2. It is known from Wiitala [15] that RB(m, m, 2) admits no CKTs. The following theorem can be regarded as an extended result of Wiitala [15]. Recall that G(m, n, r) is the knight graph of the RB(m, n, r).

**Theorem 7.** *There are no CKT on* RB(m, n, 2) *for all* n > m ≥ 5.

**Proof.** Let m and n be integers such that n > m ≥ 5. Then, there exist positive integers k and l and r, q ∈ {1, 2, 3, 4} such that m = 4k + r and n = 4l + q. Assume that there exists a CKT H on RB(m, n, 2) which is a Hamiltonian cycle on G(m, n, 2).

**Case 1:** k < l and r = q = 1. Since all vertice in {(1, 4i + 1), (m, 4i + 1)|0 ≤ i ≤ l} and {(4i + 1, 1), (4i + 1, n)|0 ≤ i ≤ k} have only 2 incident edges and we collect all incident edges from these two sets, it happens to form a cycle (1, 1), (2, 3), (1, 5), (2, 7), ... , (2, n − 2), (1, n), (3, n − 1), (5, n), (7, n − 1), ... , (m − 2, n − 1), (m, n), (m − 1, n − 2), (m, n − 4), (m − 1, n − 6), ... , (m − 1, 3), (m, 1), (m − 2, 2), (m − 4, 1), (m − 6, 2), ... , (3, 2), (1, 1), see Figure 55 for a cycle on G(13, 17, 2). This is a contradiction since this cycle does not contain all vertices of G(m, n, 2).

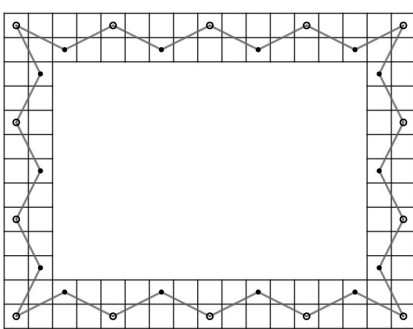

**Figure 55.** A cycle on G(13, 17, 2).

**Case 2:** k < l and r = q = 2. We obtain a contradiction similar to Case 1 by considering {(1, 4i + 1), (m − 1, 4i + 1)|0 ≤ i ≤ l} and {(4i + 1, 1), (4i + 1, n − 1)|0 ≤ i ≤ k} instead, see Figure 56a for a cycle on G(14, 18, 2).

**Case 3:** k < l and r = q = 3. We obtain a contradiction similar to Case 1 by considering {(1, 4i + 4), (m, 4i + 4)|0 ≤ i ≤ l − 1} and {(4i + 4, 1), (4i + 4, n)|0 ≤ i ≤ k − 1} instead, see Figure 56b for a cycle on G(15, 19, 2).

**Case 4:** k < l and r = q = 4. We obtain a contradiction similar to Case 1 by considering {(1, 4i + 1), (m, 4i + 4)|0 ≤ i ≤ l} and {(4i + 1, 1), (4i + 4, n)|0 ≤ i ≤ k} instead, see Figure 56c for a cycle on G(16, 20, 2).

**Case 5:** $k \leq l$, $r = 1$ and $q = 2$. We obtain a contradiction similar to Case 1 by considering $\{(1, 4i + 1), (m, 4i + 1) | 0 \leq i \leq l\}$ and $\{(4i + 1, 1), (4i + 1, n - 1) | 0 \leq i \leq k\}$ instead, see Figure 56d for a cycle on $G(13, 14, 2)$.

**Case 6:** $k \leq l$, $r = 1$ and $q = 3$. We obtain a contradiction similar to Case 1 by considering $\{(1, 4i + 2), (m, 4i + 2) | 0 \leq i \leq l\}$ and $\{(4i + 1, 2), (4i + 1, n - 1) | 0 \leq i \leq k\}$ instead, see Figure 56e for a cycle on $G(13, 15, 2)$.

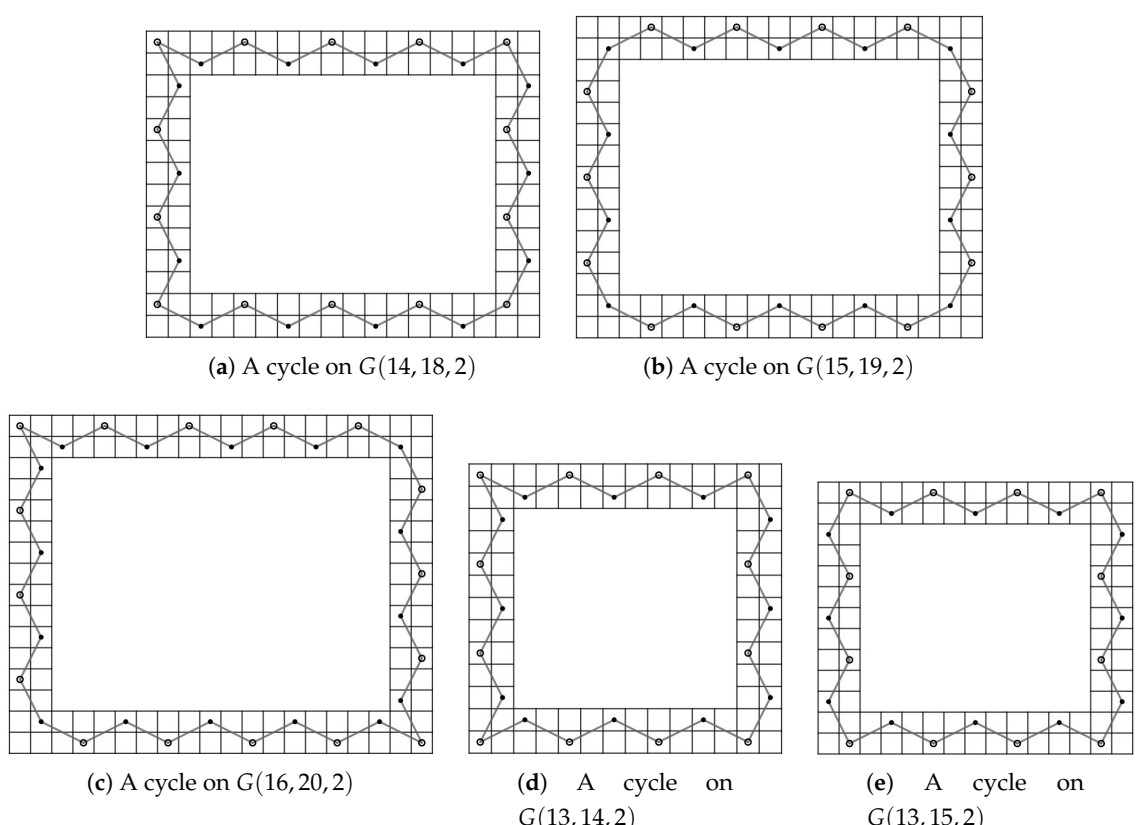

(**a**) A cycle on $G(14, 18, 2)$　　　　　　　(**b**) A cycle on $G(15, 19, 2)$

(**c**) A cycle on $G(16, 20, 2)$　　　(**d**) A cycle on $G(13, 14, 2)$　　　(**e**) A cycle on $G(13, 15, 2)$

**Figure 56.** Cycles on $G(14, 18, 2)$, $G(15, 19, 2)$, $G(16, 20, 2)$, $G(13, 14, 2)$ and $G(13, 15, 2)$, respectively.

**Case 7:** $k \leq l$, $r = 1$ and $q = 4$.

If $k = 1$, then there are some vertices (i.e., $(2, n - 4)$ and $(4, n - 4)$ which are indicated by "+" in Figure 57) that have degree more than the degree of the same vertices in the case when $k \geq 2$.

**Case 7.1:** $k = 1$. Since $(1, n)$ and $(5, n)$ have only 2 incident edges on the $G(5, n, 2)$, $(1, n) - (3, n - 1)$ and $(3, n - 1) - (5, n)$ must be in $H$ and it forces that $(1, n - 2) - (3, n - 1)$ and $(3, n - 1) - (5, n - 2)$ must not be in $H$. Then, it also forces that $(2, n - 4) - (1, n - 2)$, $(1, n - 2) - (2, n)$, $(4, n - 4) - (5, n - 2)$ and $(5, n - 2) - (4, n)$ must be in $H$. Next, since all vertice in $\{(1, 4i + 1), (5, 4i + 1) | 0 \leq i \leq l\}$, $\{(1, 4i + 2), (5, 4i + 2) | 0 \leq i \leq l - 1\}$ and $\{(2, n), (4, n)\}$ have only 2 incident edges. Collect $(2, n - 4) - (1, n - 2)$, $(1, n - 2) - (2, n)$, $(4, n - 4) - (5, n - 2)$ and $(5, n - 2) - (4, n)$ which must be in $H$ together with all incident edges from these three sets, it happen to form a cycle $(1, 1)$, $(2, 3)$, $(1, 5)$, $(2, 7)$, ... , $(1, n - 3)$, $(2, n - 1)$, $(4, n)$, $(5, n - 2)$, $(4, n - 4)$, $(5, n - 6)$, ... , $(4, 4)$, $(5, 2)$, $(3, 1)$, $(1, 2)$, $(2, 4)$, $(1, 6)$, ... , $(1, n - 6)$, $(2, n - 4)$, $(1, n - 2)$, $(2, n)$, $(4, n - 1)$, $(5, n - 3)$, ... , $(5, 5)$, $(4, 3)$, $(5, 1)$, $(3, 2)$, $(1, 1)$, see Figure 57 for a cycle on $G(5, 12, 2)$. This is a contradiction since this cycle does not contain all vertices of $G(5, n, 2)$.

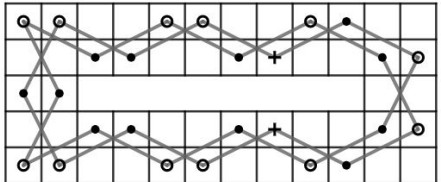

**Figure 57.** A cycle on $G(5, 12, 2)$.

**Case 7.2:** $k \geq 2$. We obtain a contradiction similar to Case 1 by considering $\{(1, 4i + 1), (1, 4i + 2), (m, 4i + 1), (m, 4i + 2) | 0 \leq i \leq l - 1\}$, $\{(1, n - 3), (2, n - 4), (m - 1, n - 4), (m, n - 3)\}$ and $\{(4i + 1, 1), (4i + 1, 2), (4i + 2, n), (4i + 4, n) | 0 \leq i \leq k - 1\}$ instead, see Figure 58 for a cycle on $G(13, 16, 2)$.

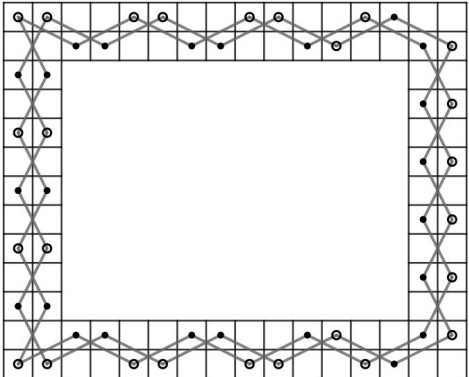

**Figure 58.** A cycle on $G(13, 16, 2)$.

**Case 8:** $k < l$, $r = 2$ and $q = 1$. We obtain a contradiction similar to Case 1 by considering $\{(1, 4i + 1), (m - 1, 4i + 1) | 0 \leq i \leq l\}$ and $\{(4i + 1, 1), (4i + 1, n) | 0 \leq i \leq k\}$ instead, see Figure 59a for a cycle on $G(14, 17, 2)$.

**Case 9:** $k \leq l$, $r = 2$ and $q = 3$. We obtain a contradiction similar to Case 1 by considering $\{(1, 4i + 2), (m - 1, 4i + 2) | 0 \leq i \leq l\}$ and $\{(4i + 1, 2), (4i + 1, n - 1) | 0 \leq i \leq k\}$ instead, see Figure 59b for a cycle on $G(14, 15, 2)$.

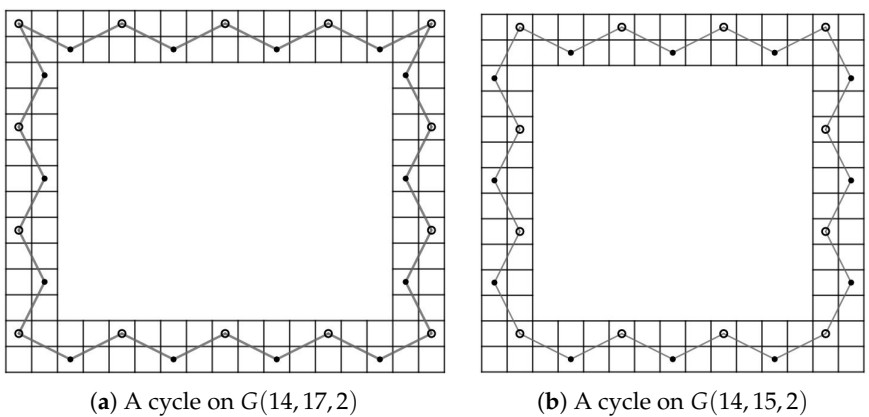

(**a**) A cycle on $G(14, 17, 2)$        (**b**) A cycle on $G(14, 15, 2)$

**Figure 59.** Cycles on $G(14, 17, 2)$ and $G(14, 15, 2)$, respectively.

**Case 10:** $k \leq l$, $r = 2$ and $q = 4$.

If $k = 1$ and $l = 1$, then it is similar to Case 7.1. We can see that $(m - 1, 5)$ (indicated by "+" in Figure 60) has degree 3 which is more than the degree of the same vertex in the case when $k \geq 2$ and $l \geq 2$.

**Case 10.1:** $k = 1$ and $l = 1$. Since $(2, 8)$ and $(6, 8)$ have only 2 incident edges on the $G(6, 8, 2)$, $(2, 8) - (4, 7)$ and $(4, 7) - (6, 8)$ must be in $H$ and it forces that $(5, 5) - (4, 7)$ must not be in $H$.

Then, it also forces that $(6,3) - (5,5)$ and $(5,5) - (6,7)$ must be in $H$. Next, since all vertice in $\{(1,1),(1,5),(2,7),(6,7),(5,1)\}$ have only 2 incident edges. Collect $(6,3) - (5,5)$ and $(5,5) - (6,7)$ which must be in $H$ and together with all incident edges from the set $\{(1,1),(1,5),(2,7),(6,7),(5,1)\}$, it happens to form a cycle $(1,1),(2,3),(1,5),(2,7),(4,8),(6,7),(5,5),(6,3),(5,1),(3,2),(1,1)$, see Figure 60. This is a contradiction since this cycle does not contain all vertices of $G(6,8,2)$.

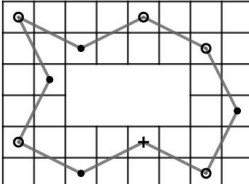

**Figure 60.** A cycle on $G(6,8,2)$.

**Case 10.2:** $k \geq 1$ and $l \geq 2$. We obtain a contradiction similar to Case 1 by considering $\{(1,4i+1)|0 \leq i \leq l\}$, $\{(4i+2,n-1),(4i+1,1)|0 \leq i \leq k\}$, $\{(m-1,5)\}$ and $\{(m,4i+3)|1 \leq i \leq l\}$ instead, see Figure 61 for a cycle on $G(14,16,2)$.

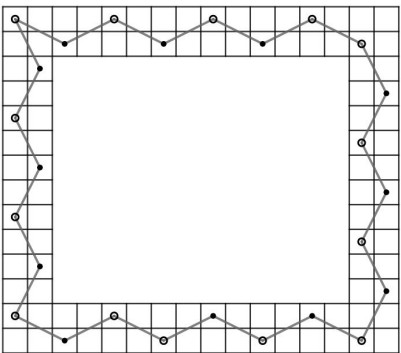

**Figure 61.** A cycle on $G(14,16,2)$.

**Case 11:** $k < l$, $r = 3$ and $q = 1$. We obtain a contradiction similar to Case 1 by considering $\{(2,4i+1),(m-1,4i+1)|0 \leq i \leq l\}$ and $\{(4i+2,1),(4i+2,n)|0 \leq i \leq k\}$ instead, see Figure 62a for a cycle on $G(15,17,2)$.

**Case 12:** $k < l$, $r = 3$ and $q = 2$. We obtain a contradiction similar to Case 1 by considering $\{(2,4i+1),(m-1,4i+1)|0 \leq i \leq l\}$ and $\{(4i+2,1),(4i+2,n-1)|0 \leq i \leq k\}$ instead, see Figure 62b for a cycle on $G(15,18,2)$.

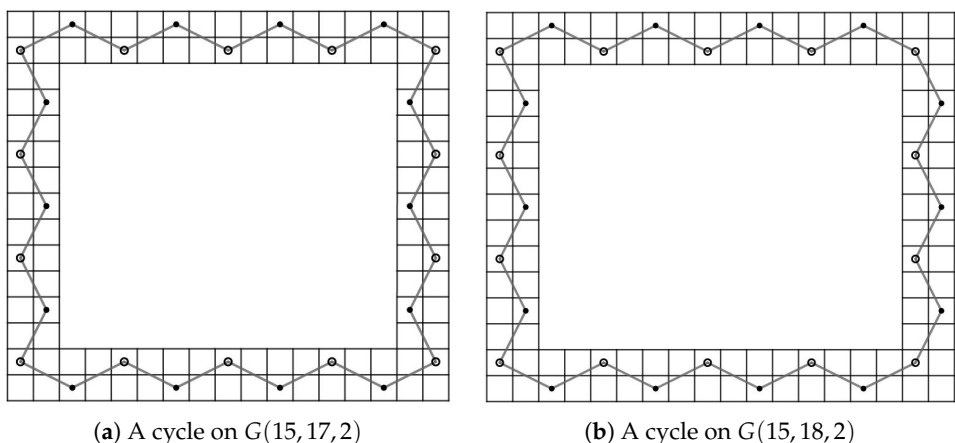

**(a)** A cycle on $G(15,17,2)$        **(b)** A cycle on $G(15,18,2)$

**Figure 62.** Cycles on $G(15,17,2)$ and $G(15,18,2)$, respectively.

**Case 13:** $k \leq l$, $r = 3$ and $q = 4$.

If $k = 1$ and $l = 1$ or $k = 1$ and $l \geq 2$, then it is similar to Case 7.1. For $k = 1$ and $l = 1$, there are some vertices (i.e., $(2,4)$, $(3,1)$, $(5,1)$ and $(6,4)$ which are indicated by "+" in Figure 63) that have a higher degree than the degree of the same vertices in the case when $k \geq 2$. For $k = 1$ and $l \geq 2$, there are some vertices (i.e., $(3,1)$ and $(5,1)$ which are indicated by "+" in Figure 64) that have degree more than the degree of the same vertices in the case when $k \geq 2$.

**Case 13.1:** $k = 1$ and $l = 1$. Since $(1,1)$, $(1,5)$, $(7,1)$ and $(7,5)$ have only 2 incident edges on $G(7,8,2)$, $(1,1) - (2,3)$, $(2,3) - (1,5)$, $(7,1) - (6,3)$ and $(6,3) - (7,5)$ must be in $H$ and it forces that $(3,1) - (2,3)$ and $(5,1) - (6,3)$ must not be in $H$. Then, it also forces that $(1,2) - (3,1)$, $(3,1) - (5,2)$, $(3,2) - (5,1)$ and $(5,1) - (7,2)$ must be in $H$. Thus, $(3,2)$ and $(5,2)$ already have two incident edges on $H$ and it forces again that $(3,2) - (2,4)$ and $(5,2) - (6,4)$ must not be in $H$. Next, since all vertice in $\{(1,1),(1,2),(1,5),(2,8),(4,8),(6,8),(7,5),(7,2),(7,1)\}$ have only 2 incident edges. Collect $(2,4) - (1,6)$, $(3,1) - (5,2)$, $(3,2) - (5,1)$ and $(6,4) - (7,6)$ which must be in $H$ together with all incident edges from $\{(1,1),(1,2),(1,5),(2,8),(4,8),(6,8),(7,5),(7,2),(7,1)\}$, it happens to form a cycle $(1,1)$, $(2,3)$, $(1,5)$, $(2,7)$, $(4,8)$, $(6,7)$, $(7,5)$, $(6,3)$, $(7,1)$, $(5,2)$, $(3,1)$, $(1,2)$, $(2,4)$, $(1,6)$, $(2,8)$, $(4,7)$, $(6,8)$, $(7,6)$, $(6,4)$, $(7,2)$, $(5,1)$, $(3,2)$, $(1,1)$, see Figure 63. This is a contradiction since this cycle does not contain all vertices of $G(7,8,2)$.

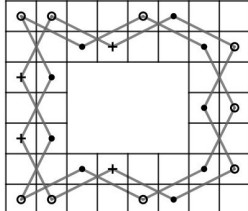

**Figure 63.** A cycle on $G(7,8,2)$.

**Case 13.2:** $k = 1$ and $l \geq 2$. Since $(1,1)$, $(1,5)$, $(7,1)$ and $(7,5)$ have only 2 incident edges on $G(7,n,2)$, $(1,1) - (2,3)$, $(2,3) - (1,5)$, $(7,1) - (6,3)$ and $(6,3) - (7,5)$ must be in $H$ and it forces that $(3,1) - (2,3)$ and $(5,1) - (6,3)$ must not be in $H$. Then, it also forces that $(1,2) - (3,1)$, $(3,1) - (5,2)$, $(3,2) - (5,1)$ and $(5,1) - (7,2)$ must be in $H$. Next, since all vertice in $\{(1,4i+1),(7,4i+1)|0 \leq i \leq l\}$, $\{(1,4i+2),(7,4i+2)|0 \leq i \leq l-1\}$ and $\{(2,n-4),(6,n-4),(2,n),(4,n),(6,n)\}$ have only 2 incident edges. Collect $(3,1) - (5,2)$ and $(3,2) - (5,1)$ which must be in $H$ together with all incident edges from these two sets, it happen to form a cycle $(1,1)$, $(2,3)$, $(1,5)$, ..., $(2,n-5)$, $(1,n-3)$, $(2,n-1)$, $(4,n)$, $(6,n-1)$, $(7,n-3)$, ..., $(7,5)$, $(6,3)$, $(7,1)$, $(5,2)$, $(3,1)$, $(1,2)$, $(2,4)$, ..., $(1,n-6)$, $(2,n-4)$, $(1,n-2)$, $(2,n)$, $(4,n-1)$, $(6,n)$, $(7,n-2)$, $(6,n-4)$, $(7,n-6)$, ..., $(6,4)$, $(7,2)$, $(5,1)$, $(3,2)$, $(1,1)$, see Figure 64 for a cycle on $G(7,16,2)$. This is a contradiction since this cycle does not contain all vertices of $G(7,4l+4,2)$.

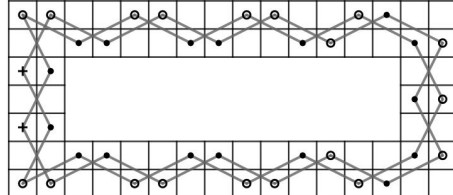

**Figure 64.** A cycle on $G(7,16,2)$.

**Case 13.3:** $k \geq 2$. We obtain a contradiction similar to Case 1 by considering $\{(1,4i+1),(m,4i+1)|0 \leq i \leq l\}$, $\{(1,4i+2),(m,4i+2)|0 \leq i \leq l-1\}$, $\{(2,n-4),(m-1,n-4)\}$, $\{(4i+2,n)|0 \leq i \leq k\}$, $\{(4i+4,n)|0 \leq i \leq k-1\}$, $\{(4i+1,1),(4i+1,2)|0 \leq i \leq k-2\}$ and $\{(4i+3,1),(4i+3,2)|1 \leq i \leq k\}$ instead, see Figure 65 for a cycle on $G(15,16,2)$.

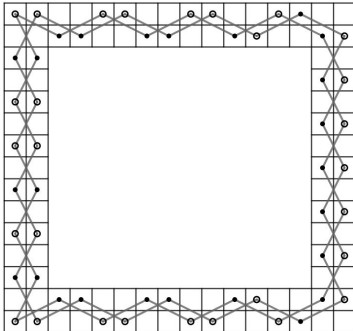

**Figure 65.** A cycle on $G(15, 16, 2)$.

**Case 14:** $k < l$, $r = 4$ and $q = 1$.

If $k = 1$, then it is similar to Case 7.1. There are some vertices (i.e., $(4, 2)$ and $(4, n - 1)$ which are indicated by "+" in Figure 66) that have degree more than the degree of the same vertices in the case when $k \geq 2$.

**Case 14.1:** $k = 1$. Since $(1, 1)$, $(1, 5)$, $(1, n - 4)$ and $(1, n)$ have only 2 incident edges on $G(8, 4l + 1, 2)$, $(1, 1) - (2, 3)$, $(2, 3) - (1, 5)$, $(1, n - 4) - (2, n - 2)$ and $(2, n - 2) - (1, n)$ must be in $H$ and it forces that $(4, 2) - (2, 3)$ and $(4, n - 1) - (2, n - 2)$ must not be in $H$. Then, it also forces that $(4, 2) - (6, 1)$ and $(4, n - 1) - (6, n)$ must be in $H$. Next, since all vertice in $\{(1, 4i + 1), (2, 4i + 1) | 0 \leq i \leq l\}$, $\{(5, 1), (5, n)\}$ and $\{(8, 4i + 2), (8, 4i + 4) | 0 \leq i \leq l - 1\}$ have only 2 incident edges. Collect $(4, 2) - (6, 1)$ and $(4, n - 1) - (6, n)$ which must be in $H$ together with all incident edges from these three sets, it happens to form a cycle $(1, 1)$, $(2, 3)$, $(1, 5)$, ..., $(1, n - 4)$, $(2, n - 2)$, $(1, n)$, $(3, n - 1)$, $(5, n)$, $(7, n - 1)$, $(8, n - 3)$, ..., $(7, 4)$, $(8, 2)$, $(6, 1)$, $(4, 2)$, $(2, 1)$, $(1, 3)$, $(2, 5)$, ..., $(2, n - 4)$, $(1, n - 2)$, $(2, n)$, $(4, n - 1)$, $(6, n)$, $(8, n - 1)$, $(7, n - 3)$, ..., $(8, 4)$, $(7, 2)$, $(5, 1)$, $(3, 2)$, $(1, 1)$, see Figure 66 for a cycle on $G(8, 13, 2)$. This is a contradiction since this cycle does not contain all vertices of $G(8, 4l + 1, 2)$.

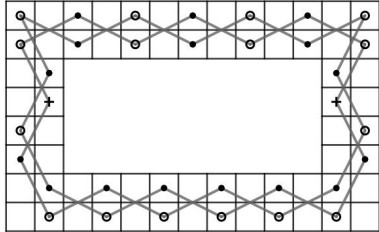

**Figure 66.** A cycle on $G(8, 13, 2)$.

**Case 14.2:** $k \geq 2$. We obtain a contradiction similar to Case 1 by considering $\{(1, 4i + 1), (2, 4i + 1) | 0 \leq i \leq l\}$, $\{(4i + 1, 1), (4i + 1, n) | 0 \leq i \leq k\}$, $\{(4i + 2, 1), (4i + 2, n) | 0 \leq i \leq k - 1\}$, $\{(m - 4, 2), (m - 4, n - 1)\}$ and $\{(m, 4i + 2), (m, 4i + 4) | 0 \leq i \leq l - 1\}$ instead, see Figure 67 for a cycle on $G(16, 17, 2)$.

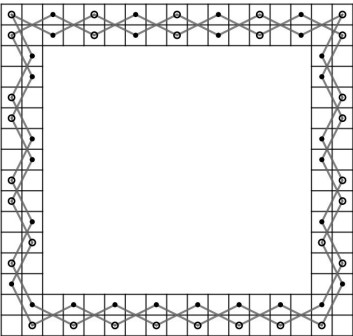

**Figure 67.** A cycle on $G(16, 17, 2)$.

**Case 15:** $k < l$, $r = 4$ and $q = 2$.

If $k = 1$ and $l \geq 2$, then it is similar to Case 7.1. The vertex $(3, n)$ (indicated by "+" in Figure 68) has degree 3 which is more than the degree of the same vertex in the case when $k \geq 2$ and $l \geq 2$.

**Case 15.1:** $k = 1$ and $l \geq 2$. Since $(1, n - 4)$ and $(1, n)$ have only 2 incident edges on $G(8, 4l + 2, 2)$, $(1, n - 4) - (2, n - 2)$ and $(2, n - 2) - (1, n)$ must be in $H$ and it forces that $(2, n - 2) - (3, n)$ must not be in $H$. Then, it also forces that $(1, n - 1) - (3, n)$ and $(3, n) - (5, n - 1)$ must be in $H$. Next, since all vertice in $\{(1, 4i + 1), (7, 4i + 2) | 0 \leq i \leq l\}$ and $\{(5, 1)\}$ have only 2 incident edges. Collect $(1, n - 1) - (3, n)$ and $(3, n) - (5, n - 1)$ which must be in $H$ together with all incident edges from these two sets, it happens to form a cycle $(1, 1)$, $(2, 3)$, $(1, 5)$, ..., $(1, n - 5)$, $(2, n - 3)$, $(1, n - 1)$, $(3, n)$, $(5, n - 1)$, $(7, n)$, $(8, n - 2)$, $(7, n - 4)$, ..., $(8, 4)$, $(7, 2)$, $(5, 1)$, $(3, 2)$, $(1, 1)$, see Figure 68 for a cycle on $G(8, 14, 2)$. This is a contradiction since this cycle does not contain all vertices of $G(8, 4l + 2, 2)$.

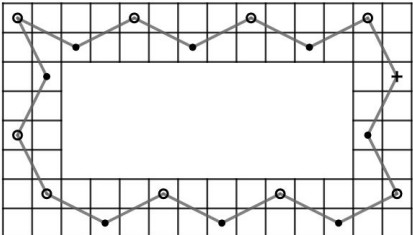

**Figure 68.** A cycle on $G(8, 14, 2)$.

**Case 15.2:** $k \geq 2$. We obtain a contradiction similar to Case 1 by considering $\{(1, 4i + 1), (m - 1, 4i + 2) | 0 \leq i \leq l\}$, $\{(4i + 1, 1) | 0 \leq i \leq k\}$, $\{(m - 5, n)\}$ and $\{(4i + 1, n - 1) | 1 \leq i \leq k - 1\}$ instead, see Figure 69 for a cycle on $G(16, 18, 2)$.

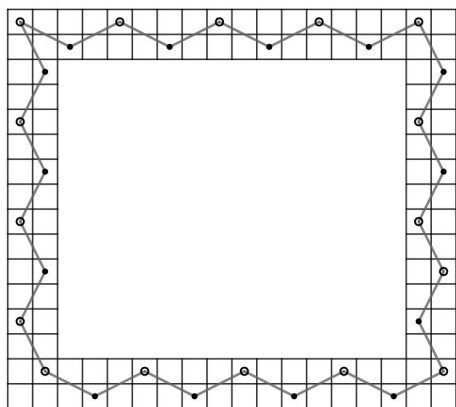

**Figure 69.** A cycle on $G(16, 18, 2)$.

**Case 16:** $k < l$, $r = 4$ and $q = 3$. We obtain a contradiction similar to Case 1 by considering $\{(1, 4i + 1), (2, 4i + 1) | 0 \leq i \leq l - 1\}$, $\{(1, n - 4), (2, n - 4)\}$, $\{(4i + 1, 1), (4i + 1, n) | 0 \leq i \leq k\}$, $\{(4i + 2, 1), (4i + 2, n) | 0 \leq i \leq k - 1\}$, $\{(m - 4, 2), (m - 4, n - 1)\}$, $\{(m, 4i + 2) | 0 \leq i \leq l\}$ and $\{(m, 4i) | 1 \leq i \leq l\}$ instead, see Figure 70 for a cycle on $G(16, 19, 2)$.

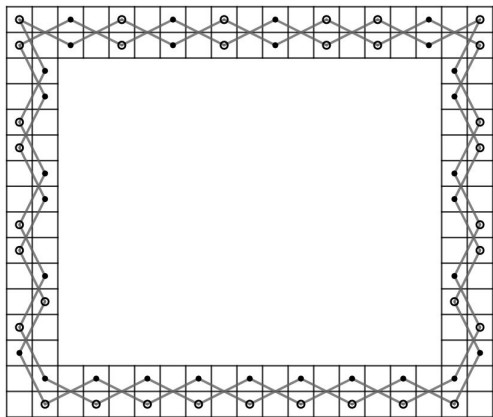

**Figure 70.** A cycle on $G(16, 19, 2)$.

This completes the proof. $\square$

Now, we are ready to prove our main theorem about the existence of a CKT on RB$(m, n, r)$.

**Theorem 8.** *An RB$(m, n, r)$ with $m, n \geq 3$ and $m, n > 2r$ has a CKT if and only if (a) $m = n = 3$ and $r = 1$ or (b) $m, n \geq 7$ and $r \geq 3$.*

**Proof.** First, for $m, n \geq 3$, $r = 1$ and $(m, n, r) \neq (3, 3, 1)$, the degree of four conner vertices of $G(m, n, 1)$ is at most one. Thus, RB$(m, n, 1)$ cannot have CKT. For $m, n \geq 5$ and $r = 2$, By the result of Wiitala[15] and Theorem 7, an RB$(m, n, 2)$ has no CKT.

Conversely, for $m = n = 3$ and $r = 1$, it is well-known that an RB$(3, 3, 1)$ has a CKT. Next, we assume that $m, n \geq 7$, $r \geq 3$ and $m, n > 2r$.

**Case 1:** $r = 3$.

**Case 1.1:** $m$ is odd and $n$ is even, or $m$ is even and $n$ is odd. We partition the RB$(m, n, 3)$ into LB$(m, n - 3, 3)$ and 7B$(m, n - 3, 3)$, see Figure 71a for RB$(10, 11, 3)$. Since $m + n - 3$ is even and $m + n - 3 \geq 12$, by Theorem 5(b), the LB$(m, n - 3, 3)$ contains an OKT from $(1, 3)$ to $(2, 2)$ and by Corollary 2(b), the 7B$(m, n - 3, 3)$ contains an OKT from $(3, 1)$ to $(2, 2)$. By joining $(1, 3)$ and $(2, 2)$ of LB$(m, n - 3, 3)$ to $(2, 2)$ and $(3, 1)$ of 7B$(m, n - 3, 3)$, respectively, we obtain a CKT on RB$(m, n, 3)$ as shown in Figure 71b for the RB$(10, 11, 3)$.

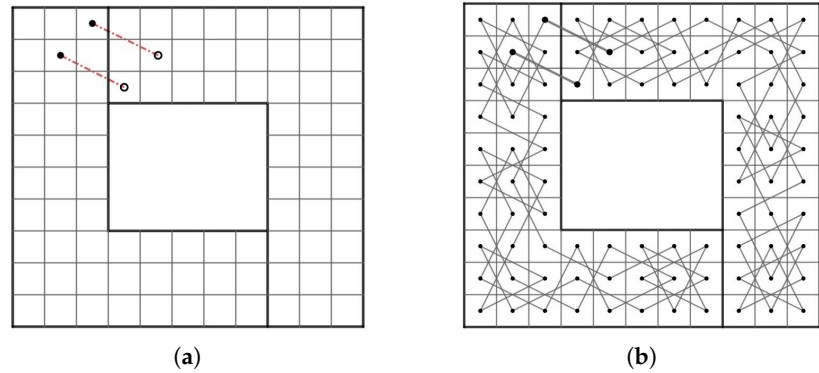

(a)          (b)

**Figure 71.** Two parts of RB$(10, 11, 3)$ and a CKT on RB$(10, 11, 3)$.

**Case 1.2:** $m$ and $n$ are odd or even. We partition the RB$(m, n, 3)$ into LB$(m, n - 3, 3)$ and 7B$(m, n - 3, 3)$, see Figure 72a for RB$(11, 13, 3)$. Since $m + n - 3$ is odd and $m + n - 3 \geq 11$, by Theorem 5(a), the LB$(m, n - 3, 3)$ contains an OKT from $(1, 2)$ to $(1, 3)$ and by Corollary 2(a), the 7B$(m, n - 3, 3)$ contains an OKT from $(2, 1)$ to $(3, 1)$. By joining $(1, 2)$ and $(1, 3)$ of LB$(m, n - 3, 3)$ to $(2, 1)$ and

$(3, 1)$ of $7B(m, n - 3, 3)$, respectively, we obtain a CKT on $RB(m, n, 3)$ as shown in Figure 72b for the $RB(11, 13, 3)$.

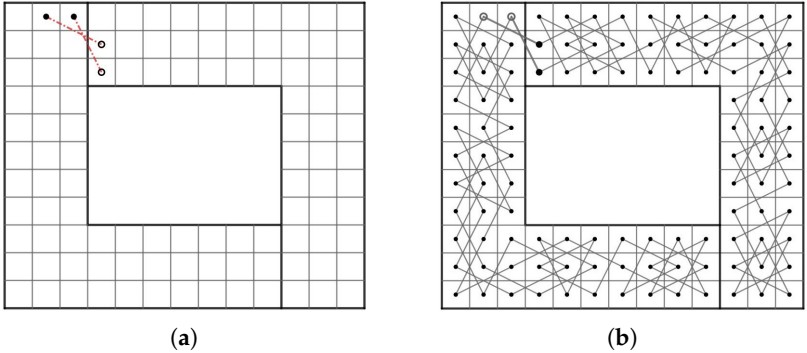

**Figure 72.** Two parts of $RB(11, 13, 3)$ and a CKT on $RB(11, 13, 3)$.

**Case 2:** for $r = 4$. We partition the $RB(m, n, 4)$ into $LB(m, n - 4, 4)$ and $7B(m, n - 4, 4)$, see Figure 73a for $RB(11, 13, 4)$. By Theorem 4 and Corollary 1, the $LB(m, n - 4, 4)$ has a CKT that contains an edge $(1, 4) - (3, 3)$ and $7B(m, n - 4, 4)$ has a CKT that contains an edge $(4, 1) - (2, 2)$. By deleting $(1, 4) - (3, 3)$ of $LB(m, n - 4, 4)$ and $(4, 1) - (2, 2)$ of $7B(m, n - 4, 4)$ and joining $(1, 4)$ and $(3, 3)$ of $LB(m, n - 4, 4)$ to $(2, 2)$ and $(4, 1)$ of $7B(m, n - 4, 4)$, respectively, we obtain a CKT on $RB(m, n, 4)$, as show in Figure 73b for $RB(11, 13, 4)$.

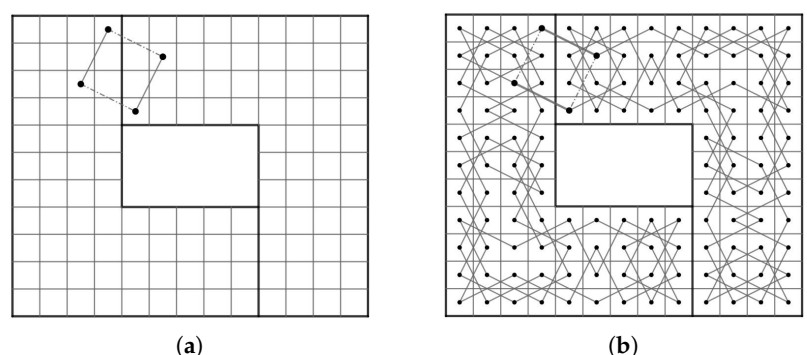

**Figure 73.** Two parts of $RB(11, 13, 3)$ and a CKT on $RB(11, 13, 3)$.

**Case 3:** $r \geq 5$.

**Case 3.1:** $r$ is even. We partition the $RB(m, n, r)$ into two $CB(r \times (n - r))$ and two $CB((m - r) \times r)$, see Figure 74a for $RB(13, 14, 6)$. There are three steps to obtain a CKT which has some edges on each partitioned board. First, we consider a $CB(r \times (m - r))$. By Theorem 1, it contains a CKT having edges $(1, m - r - 1) - (3, n - r)$ and $(r, 2) - (r - 1, 4)$. Rotate $CB(r \times (m - r))$ 90 degrees clockwise, we obtain a CKT on $CB((m - r) \times r)$ of the upper right-hand side having edges $(m - r - 1, r) - (m - r, r - 2)$ and $(2, 1) - (4, 2)$. Next, rotate $CB(r \times (m - r))$ 90 degrees counterclockwise, we obtain a CKT on $CB((m - r) \times r)$ of the lower left-hand side having edge $(m - r - 3, r - 1) - (m - r - 1, r)$. Finally, we consider a $CB(r \times (n - r))$ on the upper left-hand side. By Theorem 1, it contains a CKT having edges $(1, n - r - 1) - (3, n - r)$ and $(r, 2) - (r - 1, 4)$. Rotate $CB(r \times (n - r))$ 180 degrees clockwise, we obtain a CKT on $CB(r \times (n - r))$ of the lower right-hand side having edges $(r - 2, 1) - (r, 2)$ and $(1, n - r - 1) - (3, n - r - 3)$.

Thus, if we use the position on the $RB(m, n, r)$, there are 4 CKT on each partition having six edges, namely $(1, n - r - 1) - (3, n - r)$, $(2, n - r + 1) - (4, n - r + 2)$, $(m - r - 1, n) - (m - r, n - 2)$, $(m - r + 1, n - 1) - (m - r + 2, n - 3)$, $(m, r + 2) - (m - 2, r + 1)$ and $(m - 1, r) - (m - 3, r - 1)$.

Next, to construct a CKT on RB($m, n, r$), we delete these six edges and join these six edges: $(1, n-r-1) - (2, n-r+1)$, $(3, n-r) - (4, n-r+2)$, $(m-r-1, n) - (m-r+1, n-1)$, $(m-r, n-2) - (m-r+2, n-3)$, $(m-1, r) - (m, r+2)$ and $(m-3, r-1) - (m-2, r+1)$ instead, as shown in Figure 74b for RB($13, 14, 6$).

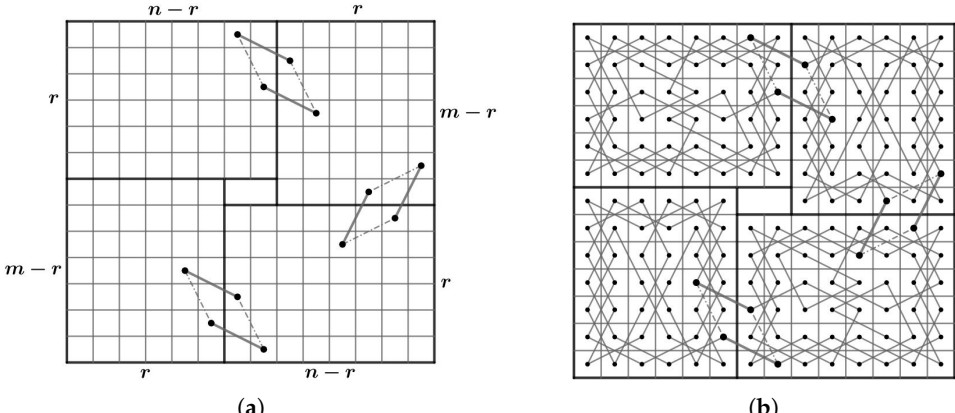

(a)　　　　(b)

**Figure 74.** Four parts of RB($13, 14, 6$) and a CKT on RB($13, 14, 6$).

**Case 3.2:** $r$ is odd. We partition the RB($m, n, r$) into two CB($r \times (n-r)$) and two CB($(m-r) \times r$), see Figure 75a for RB($12, 13, 5$). There are three steps to obtain an OKT having two end-points on each partitioned board. First, we consider a CB($r \times (m-r)$). By Theorem 6(b), it contains an OKT from $(r, 1)$ to $(2, m-r-1)$. Rotate CB($r \times (m-r)$) 90 degrees clockwise, we obtain an OKT from $(1, 1)$ to $(m-r-1, r-1)$ on CB($(m-r) \times r$) of the upper right-hand side. Next, rotate CB($r \times (m-r)$) 90 degrees counterclockwise, we obtain an OKT from $(m-r, r)$ to $(2, 2)$ on CB($(m-r) \times r$) of the lower left-hand side. Finally, we consider a CB($r \times (n-r)$) on the upper left-hand side. By Theorem 6, it contains an OKT from $(r, 1)$ to $(2, n-r-1)$. Rotate CB($r \times (n-r)$) 180 degrees clockwise, we obtain an OKT from $(1, n-r)$ and $(r-1, 2)$ on CB($r \times (n-r)$) of the lower right-hand side.

Thus, if we use the position on the RB($m, n, r$), there are 4 OKTs on each partition having eight end vertices, namely $(r, 1)$, $(2, n-r-1)$, $(1, n-r+1)$, $(m-r-1, n-1)$, $(m-r+1, n)$, $(m-1, r+2)$, $(m, r)$ and $(r+2, 2)$.

Next, to construct a CKT on the RB($m, n, r$), we join four edges: $(2, n-r-1) - (1, n-r+1)$, $(m-r-1, n-1) - (m-r+1, n)$, $(m-1, r+2) - (m, r)$ and $(r, 1) - (r+2, 2)$, as shown in Figure 75b for RB($12, 13, 5$).

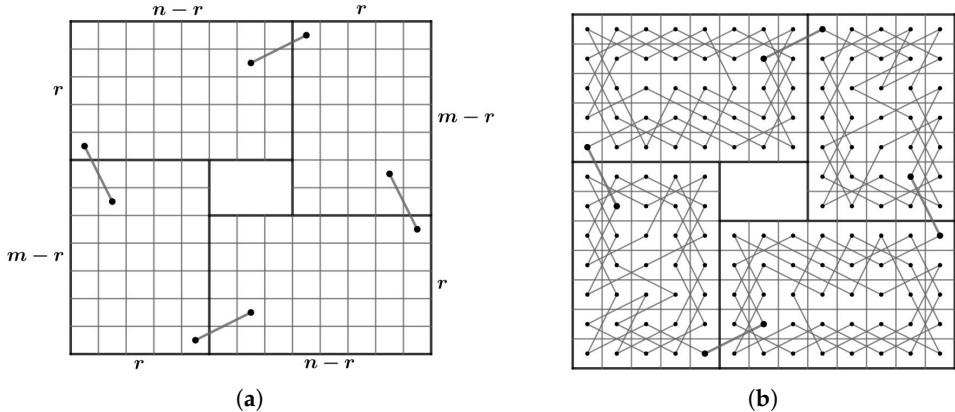

(a)　　　　(b)

**Figure 75.** Four parts of RB($12, 13, 5$) and a CKT on RB($12, 13, 5$).

This completes the proof. □

## 5. Conclusions and Discussion

In this paper, we have obtained necessary and sufficient conditions for the existence of a CKT for the RB$(m, n, r)$. In every case of Theorem 8, it can be seen that the CKTs are constructed by smaller board-pieces that have diagonal or horizontal or vertical symmetries. As a consequence, to obtain our main result, we have to study the existence of a CKT on LB$(m, n, 3)$ and LB$(m, n, 4)$. In the future, an interesting study is to find necessary and sufficient conditions for the existence of a CKT for the general L-board, namely LB$(m, n, l, u)$, which is the L-board consisting of $m$ rows $n$ with the lower leg of width $l$ and the upper leg of width $u$, see Figure 76.

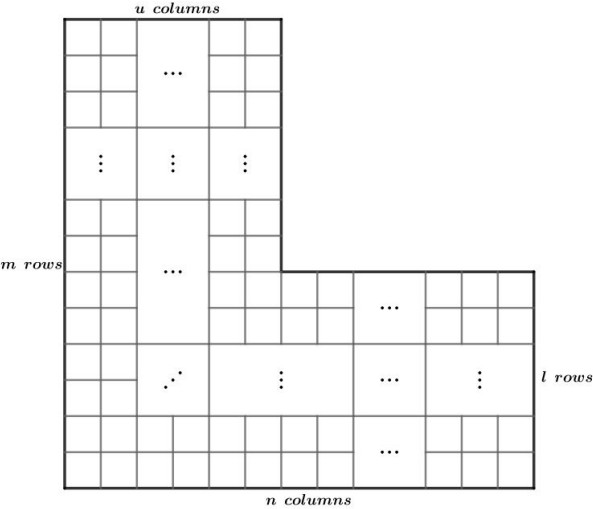

**Figure 76.** LB$(m, n, u, l)$.

**Author Contributions:** Conceptualization, W.S., R.B. and S.S.; methodology, W.S., R.B. and S.S.; validation, W.S., R.B. and S.S.; formal analysis, W.S.; investigation, W.S.; resources, W.S.; writing—original draft preparation, W.S.; writing—review and editing, R.B. and S.S.; visualization, W.S.; supervision, R.B. and S.S.; project administration, R.B. and S.S.; funding acquisition, W.S. and R.B. All authors have read and agreed to the published version of the manuscript.

**Funding:** This research was supported financially by Research Assistantship Fund, Faculty of Science, Chulalongkorn University.

**Conflicts of Interest:** The authors declare no conflict of interest.

## Abbreviations

The following abbreviations are used in this manuscript:

| | |
|---|---|
| CKT | Closed Knight's Tour |
| OKT | Open Knight's Tour |
| CB$(m \times n)$ | $m \times n$ chessboard |
| LB$(m, n, r)$ | L-board of size $(m, n, r)$ |
| 7B$(m, n, r)$ | 7-board of size $(m, n, r)$ |
| RB$(m, n, r)$ | $(m, n, r)$-ringboard |

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
