# Peer review of "Closed Knight’s Tours on (m,n,r)-Ringboards"

_symmetry, doi:10.3390/sym12081217_

Round 1
Reviewer 1 Report
I have mixed feelings about this paper. The proposed idea seems to be original and it is very interesting. Unfortunately, the quality of presentation is very low. The paper is written poorly and improperly organized, thus it is hard to follow.
The paper needs very serious improvements:
- The current state of the art should be extended. The set of references contains only 9 positions, which means that either: the overview of the literature wasn’t done properly, or, the problems shown in the paper are very marginal.
- Please clearly state the main contributions of the paper.
- Please state the scope of the paper – are the presented theorems applicable?
- Please improve the structure of the paper. It is very hard to follow the main idea. Please re-write the structure keeping in the mind “what is the main idea of the paper?”
Thank you.
Author Response
Attached, please find our response to comments of reviewer 1.

Reviewer 2 Report
======================= SUMMARY ======================
This paper is about the closed knight's tour problem on ringboards which consists in determining whether, given a ring chess board, if there exists a hamiltonian cycle in the graph where each node is a cell of the board and each edge corresponds to a move of knight (using the regular rules of chess). A ring chess board is a chess board with m lines, n columns and a rectangular hole from the cell (r + 1, r + 1) to (m - r, n - r)
The paper gives a complete dichotomy on the existence of such a cycle for every size of boards and holes that is m, n > 2r (otherwise the hole is empty) and
- m = n = 3 and r = 1
- m, n >= 7 and r >= 3
In every other case no cycle exists.
====================== MAIN COMMENT ========================
The subject is interesting and the results seems not trivial. This extends a previous 20-years-old result.
The paper is well written.
I checked all the proofs except the last cases of theorem 7 which is, from my point of view, the hardest to read.
I found one small mistake but I assume it is possible to correct it easily which is why I suggest acceptance after a minor revision.
== The mistake is Page 11 on Figure 20.
The set S has 7 cells instead of 6. As there are 8 components in (G3' - S), we do not have w(G3' - S) > |S| + 1.
== For Theorem 7, I suggest
- firstly, to replace every figure containing "three dots columns" and "three dots lines" (figuring an arbitrary number of lines and columns, for instance figures 55, 56, 57, ...) by a chessboard with a fixed value for m and n. Presently, those figures are hard to read due to the discontinuity of the drawn paths.
- secondly to explain explicitely why the argument used in the 6 first cases do not work for every cases (like case 7 or 10).
- thirdly to explain why case 7, 10, 13, ..., should be divided in two or more subcases, and why for instance, the argument for case 7.2 does not work for case 7.1.
====================== MINOR COMMENT ========================
Page 1, line 16 : a legal knights --> a legal knight's
Page 5, line 87-96 : why not using modulo notations?
Page 5, line 102 : why using s^2 + t^2 != 0 instead of s != 0 (the same notation occurs multiple times in the paper)
Page 6, line 127 : "For m = 4 and n = 5" : it was not obvious at first that it is the sole remaining case. I suggest to explicitely write this somewhere.
Page 6, line 127 : I suggest to say somewhere that FIgure9 refers to this case.
Page 8, line 138 : "to (2, 1) and contains" --> "to (2, 1) that contains"
Page 9, line 144 : "to right" --> "to the right"
Page 9, line 150 : "the followings" --> "the following two steps"
Page 13, line 206 : "4tx3" --> I suggest to use s instead of t as it is done in the rest of the paper.
Page 13, line 209 : (1, 3) and (4, 1) should be swapped ; the same occurs for (2, 2) and (4, 3).
Page 13, line 220 : I suggest to remove "If s > 0 and t = 0 then we do nothing."
Page 14, figure 29 : The two subfigures are identical.
Page 15, line 207 : "for r >= 5 is odd" --> "for r >= 5 when r is odd"
Page 16, figure 31 : This is the first figure where the "cross" notation is used to define a cell that is removed but not in S. I suggest to use the same notation every where.
Page 17, line 309 : "from (m, 1) to (2, n - 1)", I suggest to add "containing the edge (1, n) -- (3, n - 1)". Similarly, for figure 36, I suggest to explicitely say that those paths contains the edges (1, n) -- (3, n - 1) and (2, n - 1) -- (4, n). Similarly for figure 43 with the edge (1, n) -- (3, n - 1). Similarly for figure 50, with the edge (1, n) -- (3, n - 1).
Page 23, Theorem 7 : I suggest to recall the result of [9] somewhere in that page.
Page 31, case 3 of the proof of Theorem 8 : multiple times, m is used instead of n. FOr instance at line 587 "We consider CB(r x (m - r))" or line 608 (same sentence).
Page 31, line 608 : "By theorem 6" --> "By theorem 6b".
Author Response
Attached, please find our response to Reviewer 2.

Reviewer 3 Report
In "Closed Knight’s Tours on (m, n, r)-Ringboards" the authors prove the existence of closed knight's tours on 3d torus like chessboards. The proofs are through case work and appear to be correct. The value of this paper is the length to which the authors went to visualize their methods for the readers through the extensive creation of graphics. I think this merits the paper being published. The results are good enough to be accepted in this journal, but the detailed figures will be much appreciated by readers looking to prove similar results / extend this research in the future.
Author Response
Attached, please find our response to Reviewer 3.

Round 2
Reviewer 1 Report
All my comments have been properly addressed, thus I recommend acceptance of the paper.